The EMBO Journal (2013) 32, 3176–3191
www.embojournal.org

# Divergent roles of HDAC1 and HDAC2 in the regulation of epidermal development and tumorigenesis

Mircea Winter[1,6], Mirjam A Moser[1,6], Dominique Meunier[1], Carina Fischer[1,7], Georg Machat[1,8], Katharina Mattes[1], Beate M Lichtenberger[2,9], Reinhard Brunmeir[1,10], Simon Weissmann[1,11], Christina Murko[3], Christina Humer[1], Tina Meischel[1], Gerald Brosch[4], Patrick Matthias[5], Maria Sibilia[2] and Christian Seiser[1,*]

[1]Max F. Perutz Laboratories, Department of Medical Biochemistry, Vienna Biocenter, Medical University of Vienna, Vienna, Austria, [2]Department of Medicine I, Comprehensive Cancer Center, Institute for Cancer Research, Medical University of Vienna, Vienna, Austria, [3]Center for Anatomy and Cell Biology, Medical University of Vienna, Vienna, Austria, [4]Division of Molecular Biology, Biocenter, Innsbruck Medical University, Innsbruck, Austria and [5]Friedrich Miescher Institute for Biomedical Research, Novartis Research Foundation, Basel, Switzerland

The histone deacetylases HDAC1 and HDAC2 remove acetyl moieties from lysine residues of histones and other proteins and are important regulators of gene expression. By deleting different combinations of *Hdac1* and *Hdac2* alleles in the epidermis, we reveal a dosage-dependent effect of HDAC1/HDAC2 activity on epidermal proliferation and differentiation. Conditional ablation of either HDAC1 or HDAC2 in the epidermis leads to no obvious phenotype due to compensation by the upregulated paralogue. Strikingly, deletion of a single *Hdac2* allele in HDAC1 knockout mice results in severe epidermal defects, including alopecia, hyperkeratosis, hyperproliferation and spontaneous tumour formation. These mice display impaired Sin3A co-repressor complex function, increased levels of c-Myc protein, p53 expression and apoptosis in hair follicles (HFs) and misregulation of HF bulge stem cells. Surprisingly, ablation of HDAC1 but not HDAC2 in a skin tumour model leads to accelerated tumour development. Our data reveal a crucial function of HDAC1/

*Corresponding author. Max F. Perutz Laboratories, Department of Medical Biochemistry, Vienna Biocenter, Medical University of Vienna, Dr Bohr-Gasse 9/2, 1030 Vienna, Austria. Tel.: +431 4277 61770; Fax: +431 4277 9617; E-mail: christian.seiser@univie.ac.at
[6]These authors contributed equally to this work.
[7]Present address: Department of Microbiology, Tumor and Cell Biology, Karolinska Institute, Stockholm, Sweden.
[8]Present address: Department of Medicine I, Comprehensive Cancer Center, Institute for Cancer Research, Medical University of Vienna, Vienna, Austria.
[9]Present address: Division of Genetics and Molecular Medicine, Centre for Stem Cells and Regenerative Medicine, King's College London, London, UK.
[10]Present address: Singapore Institute for Clinical Science, Singapore.
[11]Present address: Biotech Research and Innovation Centre, University of Copenhagen, Copenhagen, Denmark.

HDAC2 in the control of lineage specificity and a novel role of HDAC1 as a tumour suppressor in the epidermis.
*The EMBO Journal* (2013) **32**, 3176–3191. doi:10.1038/emboj.2013.243; Published online 15 November 2013
*Subject Categories:* development; molecular biology of disease
*Keywords:* chromatin; epidermis; HDACs

## Introduction

The skin is the largest organ of the mammalian body, forming a physical barrier against the environment. The external and stratified part, the epidermis, is separated from the underlying dermis by the basement membrane and protects the body from dehydration and external impacts. The epidermal epithelium is established by differentiation of multipotent stem cells (SCs) into the interfollicular epidermis (IFE), the hair follicle (HF) and the sebaceous gland (SG) lineages. The HFs produce hair in a repeating cycle of hair growth (anagen), hair regression (catagen) and a resting phase (telogen). Lubrication of the skin surface is carried out by the SG, secreting a lipid-rich sebum. The IFE has a high regenerative potential, relying on the constant renewal of its layers—the cells are able to self-renew and give rise to the differentiating cell progeny. The bulge area, located at the HF beneath the SG harbours quiescent SCs, possessing the ability to generate all epidermal lineages (Blanpain and Fuchs, 2009). Bulge SCs also contribute to the hair cycle and to the wound repair, but under normal conditions they do not seem to be involved in the homeostasis of the epidermis (Ito *et al*, 2005). The process of terminal differentiation in the IFE is a tightly regulated process of gene expression changes ensuring the faithful layering of the IFE (Blanpain and Fuchs, 2009). Cells of the basal layer, expressing K5/K14 migrate outward, lose their mitotic potential and form the specific keratin markers in each layer. Expression of late differentiation markers like loricin, filaggrin and involucrin leads to the terminal differentiation including apoptosis and formation of the cornified envelope.

As a self-renewing tissue the epidermis is an excellent model system for studying the coordinated regulation of proliferation and differentiation (Blanpain and Fuchs, 2009). Deregulation of the pathways controlling the transition from proliferation to differentiation in the epidermis can lead to pathological responses such as skin tumours. Given that non-melanoma skin cancers such as basal cell carcinoma and squamous cell carcinoma are the most frequent human tumours, it is important to understand the basic molecular mechanisms and the contribution of epidermal SCs to skin cancerogenesis (Boehnke *et al*, 2012). Epidermal lineage commitment and morphogenesis are controlled throughout embryonic development and adulthood by an orchestrated

interplay between different signalling pathways (Blanpain and Fuchs, 2006, 2009). In addition, chromatin modifications have been shown to play a central role in epidermal morphogenesis (Aarenstrup *et al*, 2008; Ezhkova *et al*, 2009; Shaw and Martin, 2009; Ezhkova *et al*, 2011; Botchkarev *et al*, 2012; Robertson *et al*, 2012). Reversible histone acetylation is one of the best-characterized chromatin modifications and correlates with opening of local chromatin structures and transcriptional activation. Two types of enzymes, histone acetyltransferases and histone deacetylases (HDACs), control the dynamic acetylation of histones and other non-histone substrates. Eighteen mammalian HDACs have been identified and classified into four groups based on their homologies to yeast enzymes (Witt *et al*, 2009): Rpd3-like class I (HDAC1, 2, 3, 8), Hda1-like class II (HDAC4, 5, 6, 7, 9 and 10), class IV (HDAC11) HDACs and the mechanistically unrelated Sir2-like class III sirtuins (SIRT1–7). Since small molecule inhibitors of HDACs induce apoptosis, autophagy, cell-cycle arrest and differentiation in tumour cells, HDACs are attractive targets for anti-cancer treatment. Nevertheless, the precise role of individual HDACs in development and tumorigenesis is not fully understood. Thus, a fundamental understanding of the individual role of HDACs in specific cell types is therefore an essential prerequisite for the application of specific HDAC inhibitors as therapeutic drugs.

The highly homologous class I deacetylases HDAC1 and HDAC2 can homo- and hetero-dimerize and are components of the Sin3, NuRD, CoREST and NODE co-repressor complexes (reviewed in Brunmeir *et al*, 2009). Loss-of-function studies in the mouse suggest partial functional redundancy for these enzymes in different cell types and tissues (Montgomery *et al*, 2007; Yamaguchi *et al*, 2010; Chen *et al*, 2011; Jacob *et al*, 2011; Ma *et al*, 2012). Recently, LeBoeuf *et al* (2011) have shown that HDAC1 and HDAC2 play a central role in embryonic ectoderm development. Simultaneous deletion of both enzymes mediated by *K14-Cre* resulted in perinatal lethality and impairment of epidermal stratification accompanied by increased levels of p53 and impaired gene repression by the epidermal key regulator p63. Given a potential tumour maintaining role of HDAC1/2 through association with p63 in squamous cell carcinoma, HDAC1 and HDAC2 might also be promising targets in skin cancer treatment (Ramsey *et al*, 2011). By ablating different combinations of *Hdac1* and *Hdac2* alleles in the epidermis using *K5-Cre*, we revealed a specific role of HDAC1 in epidermal development and skin cancer.

## Results

### Loss of HDAC1 or HDAC2 has no obvious consequences for epidermal development

Analysis of HDAC1 and HDAC2 expression patterns in the epidermis revealed overlapping nuclear localization in the basal, suprabasal and differentiating cell layers in the murine epidermis and HFs (Supplementary Figures S1A and B). To examine potential regulatory functions of HDAC1 and HDAC2 in epidermal development, we crossed mice with floxed *Hdac1* or *Hdac2* alleles (Yamaguchi *et al*, 2010) to a *K5-Cre* mouse line (Ramirez *et al*, 2004). For simplicity, *Hdac1*$^{f/f}$ *K5-Cre* mice and *Hdac2*$^{f/f}$ *K5-Cre* mice are referred to as *Hdac1*$^{\Delta/\Delta ep}$ and *Hdac2*$^{\Delta/\Delta ep}$ mice. In agreement with

previously published data (LeBoeuf *et al*, 2011), *Hdac1*$^{\Delta/\Delta ep}$ and *Hdac2*$^{\Delta/\Delta ep}$ mice showed complete loss of the respective deacetylase, and were viable, fertile and displayed normal epidermal and HF development (Supplementary Figures S1C, D and J). In accordance with an unaltered epidermal thickness, immunofluorescence (IF) staining of mouse back skin sections revealed comparable expression of the proliferation marker Ki67 in *Cre*-deficient control, *Hdac1*$^{\Delta/\Delta ep}$ and *Hdac2*$^{\Delta/\Delta ep}$ mice (Supplementary Figures S1E and F). Remarkably, 24% of *Hdac1*$^{\Delta/\Delta ep}$ mice developed hyperproliferative scars on their tails, while no scars were observed on the tails of *Hdac2*$^{\Delta/\Delta ep}$ mice (Supplementary Figure S1G). The scar tissue showed a significant increase in proliferation as shown by quantification of the Ki67 immunohistochemical (IHC) staining (Supplementary Figure S1H).

HDAC2 protein levels were up-regulated in the epidermis of *Hdac1*$^{\Delta/\Delta ep}$ mice and HDAC1 expression was increased in the *Hdac2*$^{\Delta/\Delta ep}$ epidermis, while mRNA levels were unchanged, indicating a post-transcriptional compensatory mechanism (Supplementary Figures S1I and J). To estimate the contribution of these two HDACs to the overall HDAC activity, we performed HDAC activity assays with epidermal protein extracts and tritium-acetate labelled histones as a substrate. These experiments revealed no significant changes in total cellular HDAC activity in the absence of either HDAC1 or HDAC2 (Supplementary Figure S1K).

These data suggest normal epidermal development in the absence of either HDAC1 or HDAC2 with a higher incidence of scar formation on HDAC1-deficient epidermis.

### Severe developmental abnormalities in the epidermis of *Hdac1*$^{\Delta/\Delta ep}$ *Hdac2*$^{\Delta/+ ep}$ mice

To investigate redundant and non-redundant functions of HDAC1 and HDAC2, we generated mice with combined ablations of *Hdac1* and *Hdac2* alleles. As reported by LeBoeuf *et al* (2011), mice with simultaneous deletion of *Hdac1* and *Hdac2* (*Hdac1*$^{\Delta/\Delta ep}$ *Hdac2*$^{\Delta/\Delta ep}$) were not viable confirming an indispensable role of HDAC1 and HDAC2 for epidermal development (Supplementary Figure S1L).

HDAC2-deficient mice with a single *Hdac1* allele (*Hdac1*$^{\Delta/+ ep}$ *Hdac2*$^{\Delta/\Delta ep}$) and compound heterozygous mice (*Hdac1*$^{\Delta/+ ep}$ *Hdac2*$^{\Delta/+ ep}$) showed normal development and were indistinguishable from *Cre*-deficient control mice (Figure 1A; Supplementary Figure S1C). Interestingly, deletion of one *Hdac2* allele in *Hdac1*$^{\Delta/\Delta ep}$ mice (*Hdac1*$^{\Delta/\Delta ep}$ *Hdac2*$^{\Delta/+ ep}$) displayed severe developmental abnormalities (Figures 1A and C). Compared to their control littermates, *Hdac1*$^{\Delta/\Delta ep}$ *Hdac2*$^{\Delta/+ ep}$ mice were smaller shortly after birth, gained less weight from P3 onward and had reduced weight during adulthood (Figures 1B and C). *Hdac1*$^{\Delta/\Delta ep}$ *Hdac2*$^{\Delta/+ ep}$ mice exhibited progressive alopecia and adult animals had little body hair, shorter whiskers and scaly tail regions (Figures 1C and E). Remarkably, older *Hdac1*$^{\Delta/\Delta ep}$ *Hdac2*$^{\Delta/+ ep}$ mice (>4 months) but not *Hdac1*$^{\Delta/\Delta ep}$ or *Hdac1*$^{\Delta/+ ep}$ *Hdac2*$^{\Delta/\Delta ep}$ mice spontaneously developed papilloma-like lesions on the skin of the tail, back, head or extremities (Figure 1D). IHC analysis showed that HDAC2 was expressed in all layers of *Hdac1*$^{\Delta/\Delta ep}$ *Hdac2*$^{\Delta/+ ep}$ epidermis indicating that the developmental defects of *Hdac1*$^{\Delta/\Delta ep}$ *Hdac2*$^{\Delta/+ ep}$ mice were not due to restricted HDAC2 expression patterns (Supplementary Figure S2A).

Deletion of one *Hdac1* allele in *Hdac2*$^{\Delta/\Delta ep}$ epidermis or one *Hdac2* allele in *Hdac1*$^{\Delta/\Delta ep}$ epidermis led to the

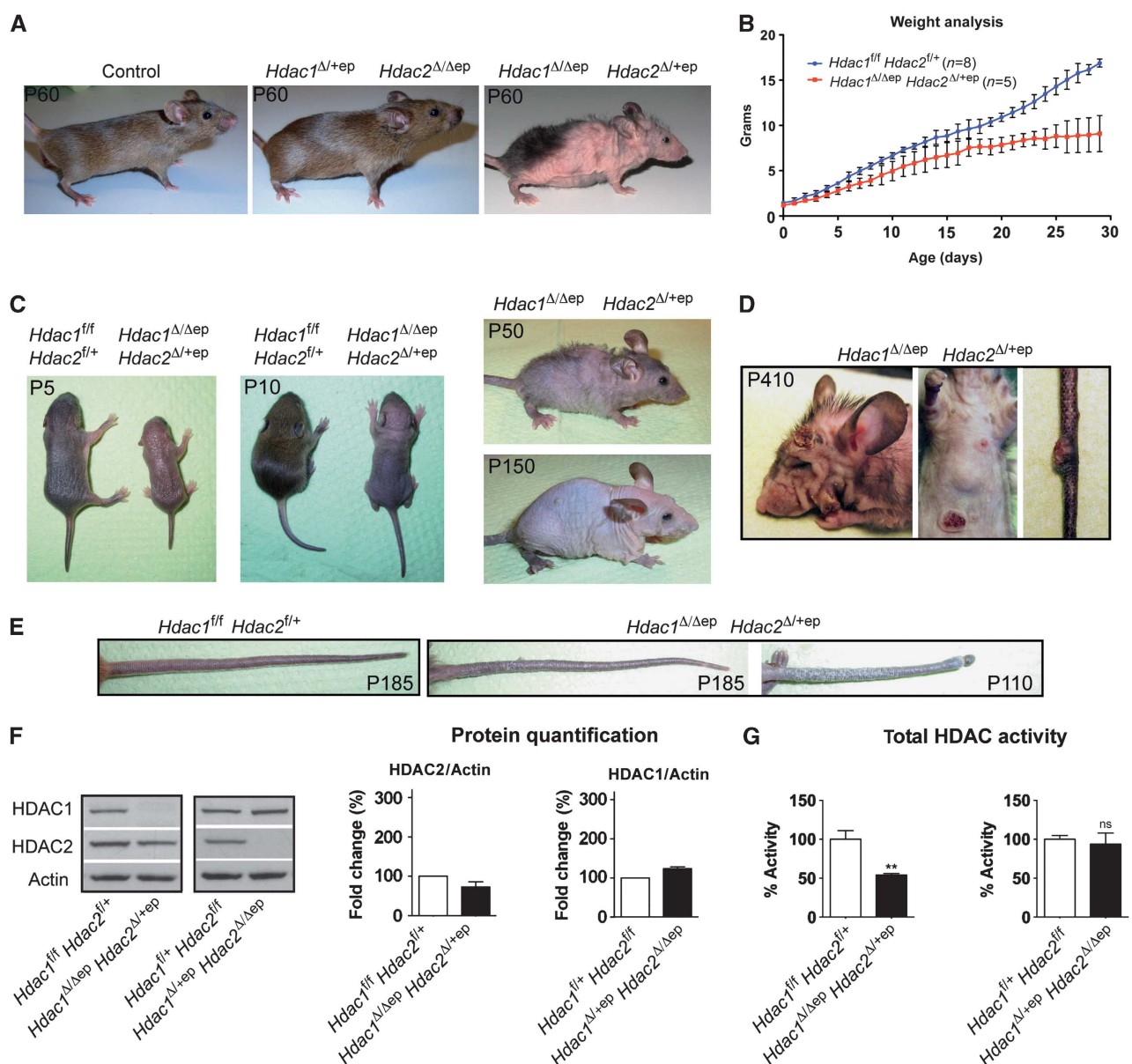

**Figure 1** Severe phenotype of $Hdac1^{\Delta/\Delta ep}\ Hdac2^{\Delta/+ep}$ mice. (**A**) Pictures of adult wild-type control, $Hdac1^{\Delta/+ep}\ Hdac2^{\Delta/\Delta ep}$ and $Hdac1^{\Delta/\Delta ep}$ $Hdac2^{\Delta/+ep}$ mice. (**B**) Weight curve of control ($n=8$) and $Hdac1^{\Delta/\Delta ep}\ Hdac2^{\Delta/+ep}$ ($n=5$) mice from P1 to P30. (**C**) Progressive alopecia of $Hdac1^{\Delta/\Delta ep}\ Hdac2^{\Delta/+ep}$ mice with age. (**D**) Spontaneous skin tumour formation in older $Hdac1^{\Delta/\Delta ep}\ Hdac2^{\Delta/+ep}$ mice. (**E**) Tails of adult control and two individual $Hdac1^{\Delta/\Delta ep}\ Hdac2^{\Delta/+ep}$ mice (P185, P185, P110) are shown. (**F**) Immunoblot analysis of tail epidermal extracts with antibodies against HDAC1, HDAC2, and β-actin as a loading control. Immunoblot signals quantified by densitometric scanning are shown relative to the β-actin signal. $n=4$. (**G**) Total HDAC activity of epidermal extracts from control, $Hdac1^{\Delta/\Delta ep}\ Hdac2^{\Delta/+ep}$ and $Hdac1^{\Delta/+ep}$ $Hdac2^{\Delta/\Delta ep}$ mice, shown in %. $n=4$. $P=0.0022$ and $0.5172$. (**F, G**) Data are mean ± s.d. Source data for this figure is available on the online supplementary information page.

corresponding reduction in mRNA levels (Supplementary Figure S2B). However, immunoblot analysis revealed only small changes in the amounts of HDAC1 and HDAC2 protein in the epidermis of $Hdac1^{\Delta/\Delta ep}\ Hdac2^{\Delta/+ep}$ and $Hdac1^{\Delta/+ep}$ $Hdac2^{\Delta/\Delta ep}$ mice compared to littermate controls (Figure 1F), indicating that the regulatory cross-talk between HDAC1 and HDAC2 was also functional in these mice. The total cellular HDAC activity in epidermal extracts was significantly reduced in $Hdac1^{\Delta/\Delta ep}\ Hdac2^{\Delta/+ep}$ but not in $Hdac1^{\Delta/+ep}\ Hdac2^{\Delta/\Delta ep}$ mice (Figure 1G). Accordingly, different histone acetylation marks including H3K4, H3K27, H3K56 and H4K8 were specifically increased in $Hdac1^{\Delta/\Delta ep}\ Hdac2^{\Delta/+ep}$ epidermis (Supplementary Figure S2C). These data demonstrate that a

single *Hdac1* allele can compensate for the loss of HDAC2 in the epidermis, whereas a single *Hdac2* allele fails to balance for HDAC1 deficiency.

To ask whether this difference was due to different expression levels of HDAC1 and HDAC2 in keratinocytes, immunoblot signals for HDAC1 and HDAC2 were calibrated with recombinant proteins and measured. As shown in Supplementary Figure S2D, the expression of HDAC1 and HDAC2 protein was very similar as measured with two different antibodies for each enzyme. These findings suggest that HDAC1 and HDAC2 have only partially redundant functions with a predominant role of HDAC1 in epidermal development.

### *Hdac1*$^{\Delta/\Delta ep}$ *Hdac2*$^{\Delta/+ep}$ mice display disturbed HF development

One obvious characteristic in the *Hdac1*$^{\Delta/\Delta ep}$ *Hdac2*$^{\Delta/+ep}$ mice was the reduced pelage and the progressive loss of the remaining hair during adulthood (Figures 1A and C). Therefore, we examined the hair development in more detail in the back skin of these mice during HF morphogenesis (P5–P10), telogen (P18) and anagen (P35) by haematoxylin and eosin (H&E) stainings (Figure 2A). The HF number of control and mutant mice was comparable (Supplementary Figure S3A). However, already during HF morphogenesis (P5

and P10) mutant HFs were shorter and disordered compared to *Hdac1*$^{f/f}$ *Hdac2*$^{f/+}$ control mice (Figure 2A and C). At P18 *Hdac1*$^{\Delta/\Delta ep}$*Hdac2*$^{\Delta/+ep}$, HFs were significantly longer and failed to enter the telogen phase in a synchronized manner. Later, the mutant HF became atrophied, lost their morphology and could not enter the anagen phase at P35 as the HFs of wild-type littermates (Figures 2A and B). In line with the impaired ability of the HF to fully develop during morphogenesis we detected an increase in p53 expression and apoptosis as detected by cleaved caspase-3 staining in mutant HFs at P10 (Figure 2D). Thus, p53-induced apoptosis may

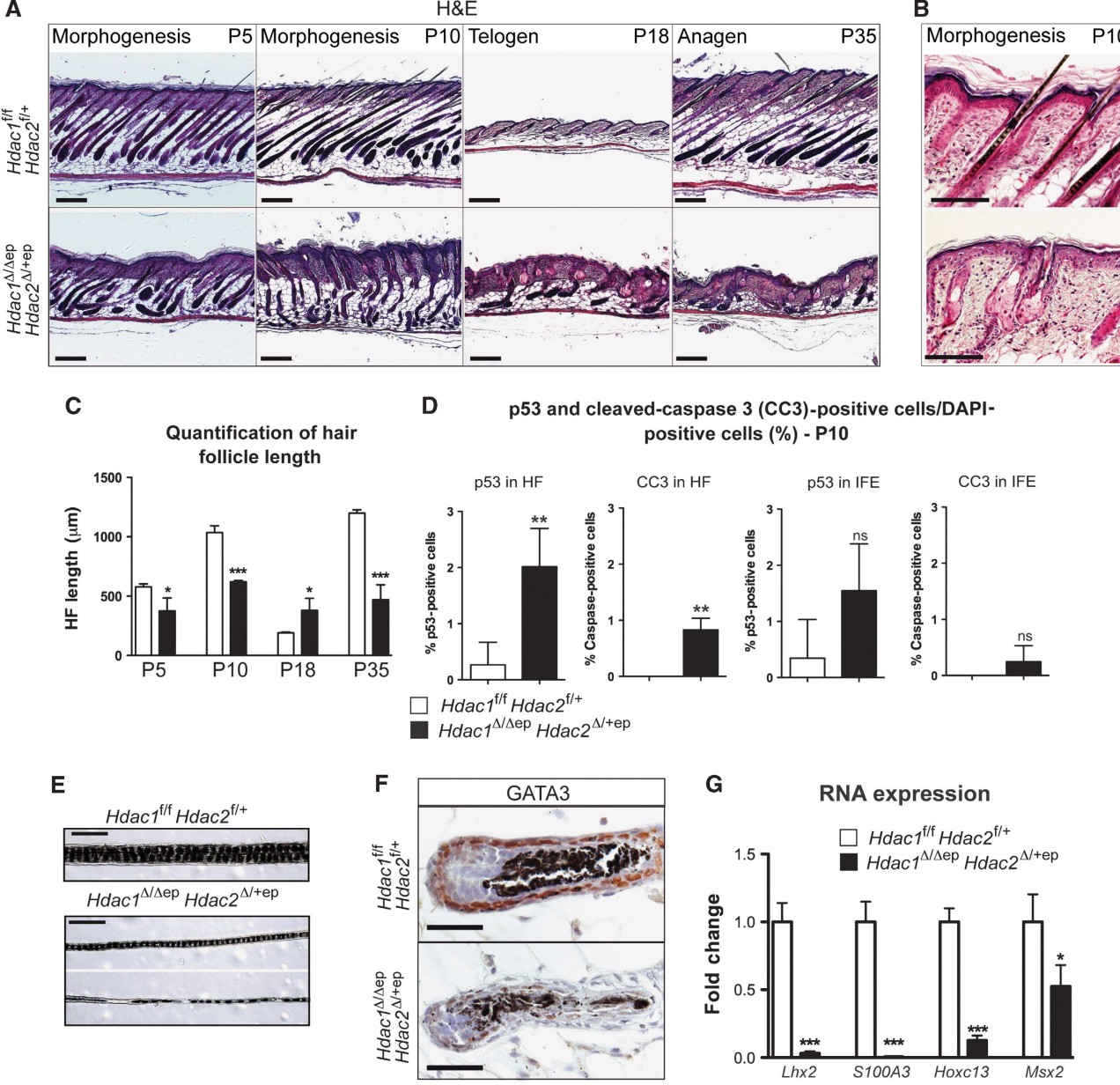

**Figure 2** Progressive hair loss and deregulated HF development in *Hdac1*$^{\Delta/\Delta ep}$ *Hdac2*$^{\Delta/+ep}$ mice. (**A**) H&E staining of back skin sections from control and *Hdac1*$^{\Delta/\Delta ep}$ *Hdac2*$^{\Delta/+ep}$ mice at P5, P10, P18 and P35. Scale bar: 200 µm. (**B**) Higher magnification of H&E staining of back skin sections from control and *Hdac1*$^{\Delta/\Delta ep}$ *Hdac2*$^{\Delta/+ep}$ mice at P10. Scale bar: 200 µm. (**C**) Quantification of the hair follicle length of back skin hair from control and *Hdac1*$^{\Delta/\Delta ep}$ *Hdac2*$^{\Delta/+ep}$ littermates at different stages. $n = 3$; $P = 0.0353, 0.0002, 0.0336$ and $0.0006$. (**D**) p53 and cleaved caspase 3-positive (CC3) cells as a fraction of DAPI-stained cells in the HF and in the IFE. $n = 4$; $P = 0.0045, 0.001, 0.0690$ and $0.2060$. (**E**) Brightfield image of back skin hair from littermate control and *Hdac1*$^{\Delta/\Delta ep}$ *Hdac2*$^{\Delta/+ep}$ mice (P35). Scale bar: 100 µm. (**F**) Back skin HF (anagen, P10) of control and *Hdac1*$^{\Delta/\Delta ep}$ *Hdac2*$^{\Delta/+ep}$ littermates was stained with an antibody against GATA3. Nuclei were counterstained with Mayer's hemalaun. Scale bar: 50 µm. (**G**) Relative mRNA expression of genes important for hair development. $n = 3–4$, $P = 0.001, 0.0003, 0.0001$ and $0.0326$. (**C, D, G**) Data represent means ± s.d.

considerably contribute to HF degeneration in $Hdac1^{\Delta/\Delta ep}$ $Hdac2^{\Delta/+ep}$ mice. Remarkably, in comparison to the HFs the $Hdac1^{\Delta/\Delta ep}Hdac2^{\Delta/+ep}$ IFE seemed to be more resistant to p53-induced apoptosis (Figure 2D; Supplementary Figures S3B and C). $Hdac1^{\Delta/\Delta ep}$ $Hdac2^{\Delta/+ep}$ hair was also thinner and the medulla segmentation was partly disrupted (Figure 2E). IHC staining for the transcription factor GATA3, which is crucial for epidermal lineage determination and differentiation of the IRS (Kaufman *et al*, 2003), revealed loss of the protein in $Hdac1^{\Delta/\Delta ep}$ $Hdac2^{\Delta/+ep}$ HFs confirming an impaired hair morphogenesis programme (Figure 2F). Lhx2, a master regulator of HF SCs (Folgueras *et al*, 2013) was also down-regulated in HFs of $Hdac1^{\Delta/\Delta ep}$ $Hdac2^{\Delta/+ep}$ mice (Figure 2G). Microarray gene expression analysis of control and $Hdac1^{\Delta/\Delta ep}$ $Hdac2^{\Delta/+ep}$ epidermis revealed that the disturbed hair development was accompanied by deregulation of several genes implicated in hair formation, HF development and hair cycle (Supplementary Table S1). Among these, several hair shaft keratins (*Krt31–Krt86*) and keratin-associated proteins (*Krtap1–Krtap25*) as well as the S100a3 protein, an important structural component of the hair cuticle (Kizawa *et al*, 1998), were found to be strongly diminished in their expression in mutant epidermis (Figure 2G). Similarly, expression of the important regulators of hair development *Hoxc13* and *Msx2* was significantly reduced. In summary, hair development was severely impaired due to disturbed HF morphogenesis and a failure to properly enter the hair cycle linked to increased p53 expression and apoptosis resulting in degeneration of $Hdac1^{\Delta/\Delta ep}$ $Hdac2^{\Delta/+ep}$ HFs.

### Hyperkeratosis in $Hdac1^{\Delta/\Delta ep}$ $Hdac2^{\Delta/+ep}$ mice

H&E staining and IF analysis of K1, K14 and the cornification marker involucrin revealed an enlargement of the basal layer, spinous layer and stratum corneum in the epidermis of $Hdac1^{\Delta/\Delta ep}$ $Hdac2^{\Delta/+ep}$ epidermis (Figures 3A, B and D). Small proline rich (Sprr) proteins and late cornified envelope proteins (LCEs) are important for the process of cornification and contribute to barrier function formation (Martin *et al*, 2004). Sprr and LCE protein families are encoded by gene clusters, which as part of the epidermal differentiation complex (EDC) are often coordinately expressed and regulated. Several members of the Sprr and LCE protein families were found up-regulated in $Hdac1^{\Delta/\Delta ep}$ $Hdac2^{\Delta/+ep}$ mice by qRT-PCR analyses and gene expression profiling of epidermal mRNA (Figure 3C; Supplementary Table S1). This up-regulation is consistent with the hyperkeratosis observed in $Hdac1^{\Delta/\Delta ep}$ $Hdac2^{\Delta/+ep}$ epidermis. The observed hyperkeratosis and epidermal hyperproliferation was not connected to a defect in barrier function, since both transe-

pidermal water loss (TEWL) test and toluidine blue dye exclusion assay gave no significant difference between $Hdac1^{\Delta/\Delta ep}$ $Hdac2^{\Delta/+ep}$ mice and wild-type littermates. Also skin barrier genes were unaffected in the mutant epidermis (Supplementary Figures S4A–C). These results show a deregulation of differentiation-related genes and hyperkeratosis in the epidermis of $Hdac1^{\Delta/\Delta ep}$ $Hdac2^{\Delta/+ep}$ mice.

### Increased SGs and IFE hyperproliferation in $Hdac1^{\Delta/\Delta ep}$ $Hdac2^{\Delta/+ep}$ mice

In order to survey the SG lineage, we stained control and $Hdac1^{\Delta/\Delta ep}$ $Hdac2^{\Delta/+ep}$ skin whole mounts with Oil Red O, a histochemical dye specific for lipid-containing cells including sebocytes. $Hdac1^{\Delta/\Delta ep}$ $Hdac2^{\Delta/+ep}$ SGs were significantly enlarged and filled the region of the HF shaft, which was underdeveloped or entirely absent (Figure 3E). The serine protease kallikrein related-peptidase 6 (Klk6/BSSP), which is predominantly expressed in the SGs (Frye *et al*, 2003; Komatsu *et al*, 2005), was up-regulated in the epidermis of $Hdac1^{\Delta/\Delta ep}$ $Hdac2^{\Delta/+ep}$ mice (Figure 3F).

To determine whether the observed thickening of the epidermis in $Hdac1^{\Delta/\Delta ep}$ $Hdac2^{\Delta/+ep}$ mice was due to epidermal hyperproliferation, we analysed BrdU incorporation at P5 and Ki67 expression at P30. Both were found to be up-regulated in the IFE of $Hdac1^{\Delta/\Delta ep}$ $Hdac2^{\Delta/+ep}$ mice (Figures 3H and I; Supplementary Figure S3D). We further detected a significant induction of epithelial mitogen (*Epgn*) and adenosine deaminase (*Ada*), two genes crucial for epithelial morphogenesis and proliferation, in the mutant epidermis by qRT-PCR (Figure 3G). Keratin 6 (K6), a marker for epidermal hyperproliferation is, under normal conditions, predominantly found in the companion layer of HFs and becomes expressed in the epidermis only upon stress or injury (Mazzalupo *et al*, 2003). Notably, we found up-regulation of K6 as well as other stress-associated genes including keratin 16 (*Krt16*), *S100a8* and *S100a9* (Supplementary Figure S3E; Supplementary Table S1). However, FACS analysis for markers of immune cells at P5 indicated no significant increase in immune infiltrates in the $Hdac1^{\Delta/\Delta ep}$ $Hdac2^{\Delta/+ep}$ epidermis excluding that an inflammatory condition could be responsible for the observed hyperproliferation (Supplementary Figures S4D and E). These data show that haploinsufficiency of *Hdac2* in the absence of HDAC1 resulted in increased SGs and hyperproliferation of the IFE.

### Changes in lineage determination in $Hdac1^{\Delta/\Delta ep}$ $Hdac2^{\Delta/+ep}$ epidermis

Since all three epidermal lineages were affected in their development in $Hdac1^{\Delta/\Delta ep}$ $Hdac2^{\Delta/+ep}$ mice, we examined

**Figure 3** $Hdac1^{\Delta/\Delta ep}$ $Hdac2^{\Delta/+ep}$ mice display impaired epidermal development and hyperkeratosis. (**A**) IF analysis of back skin sections of adult control, $Hdac1^{\Delta/\Delta ep}$ $Hdac2^{\Delta/+ep}$ and $Hdac1^{\Delta/+ep}$ $Hdac2^{\Delta/\Delta ep}$ mice (P35) with antibodies specific for K14 and K1. Nuclei were counterstained with DAPI. Scale bar: 50 μm. (**B**) Quantification of epidermal thickness as shown in (**A**) in littermate controls and $Hdac1^{\Delta/\Delta ep}$ $Hdac2^{\Delta/+ep}$ mice. $n = 3$. $P = 0.0150$. (**C**) qRT–PCR analysis of mRNA expression of *Sprr* genes in the epidermis from $Hdac1^{\Delta/\Delta ep}$ $Hdac2^{\Delta/+ep}$ and control littermates. $n = 3$. $P = 0.0190$ and 0.0139. (**D**) IHC analysis of control and $Hdac1^{\Delta/\Delta ep}$ $Hdac2^{\Delta/+ep}$ back skin sections for involucrin (P35). The nuclei were counterstained with Mayer's hemalaun. Scale bar: 100 μm. (**E**) Brightfield images of Oil red O and haematoxylin-stained skin whole mounts of control and $Hdac1^{\Delta/\Delta ep}$ $Hdac2^{\Delta/+ep}$ mice (P90). Dashed line marks SG, straight line marks HF unit. Scale bar: 200 μm. SG areas for individual HF were quantified and shown on the right panel. $n = 30$. $P = 0.0315$. (**F**) Relative mRNA expression of the *kallikrein-related peptidase 6*. $n = 3$. $P = 0.0165$. (**G**) qRT–PCR analyses of *Epgn* and *Ada* in the epidermis of control and $Hdac1^{\Delta/\Delta ep}$ $Hdac2^{\Delta/+ep}$ mice. $n = 3$. $P = 0.0122$ and 0.0250. (**H**) Quantification of IF signals for BrdU in HF (left panel) and IFE (right panel) of control and $Hdac1^{\Delta/\Delta ep}$ $Hdac2^{\Delta/+ep}$ mice (P5) after 2 h BrdU labelling. $n = 4$. $P = 0.0007$ and 0.0068. (**I**) IHC analysis for the proliferation antigen Ki67 in control and $Hdac1^{\Delta/\Delta ep}$ $Hdac2^{\Delta/+ep}$ tail skin (P30). Scale bar: 100 μm. Quantification of Ki67-positive cells in tail epidermis is shown on the right. $n = 3$. $P = 0.0150$. (**B–I**) Mean value with s.d. is shown.

the expression of markers for SCs located in the HF bulge (*CD34, Keratin 15*) and hair sheath (*Sox9, Lgr5, Lgr6*). To this end, we performed qRT–PCR analyses of the epidermis of $Hdac1^{\Delta/\Delta ep}$ $Hdac2^{\Delta/+ep}$ mice and control littermates. Strikingly, most analysed SC genes such as *CD34, Keratin 15, Lgr5* and *Sox9* were significantly down-regulated in adult $Hdac1^{\Delta/\Delta ep}$ $Hdac2^{\Delta/+ep}$ epidermis (Figure 4A). The

R-spondin receptor Lgr5 is a marker for multipotent SCs able to generate all lineages of the HF (Schuijers and Clevers, 2012). To investigate its local expression pattern, we performed *in situ* hybridization experiments. In control epidermis, *Lgr5* was located in the HF bulb and ORS, whereas Lgr5 was not detectable in HFs of $Hdac1^{\Delta/\Delta ep}$ $Hdac2^{\Delta/+ep}$ mice (Supplementary Figure S5A). On the

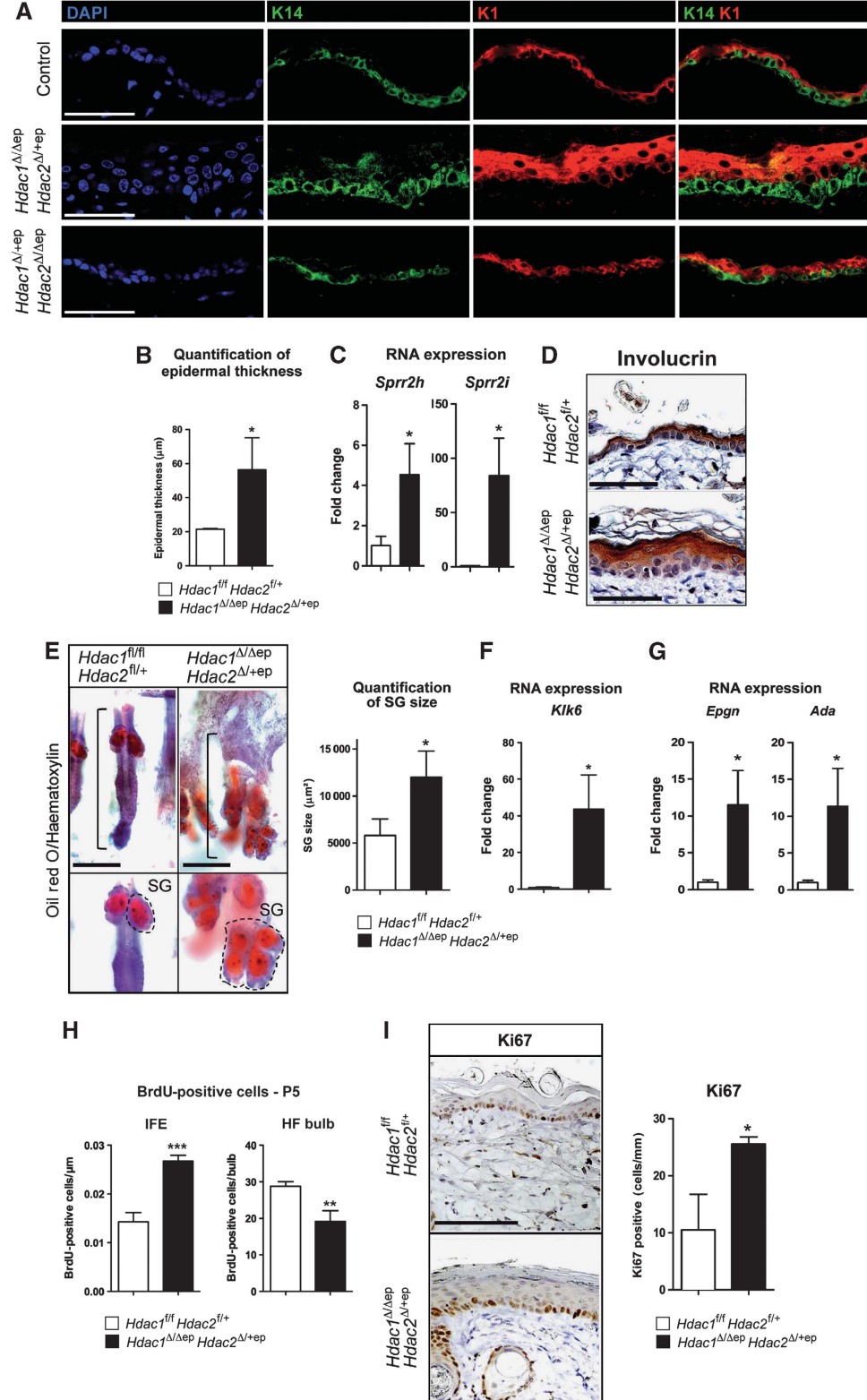

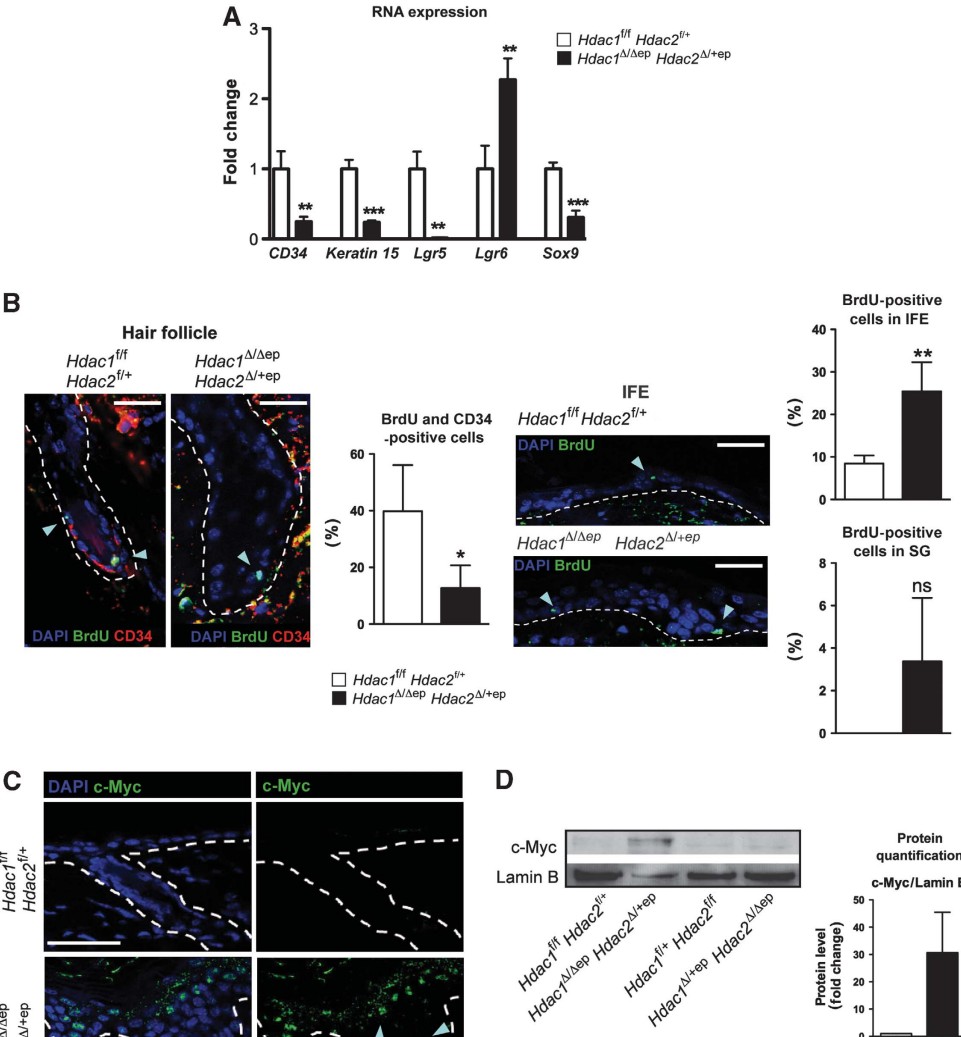

**Figure 4** Epidermal hyperproliferation, deregulated epidermal lineage commitment and c-Myc activation in $Hdac1^{\Delta/\Delta ep}$ $Hdac2^{\Delta/+ep}$ mice. (**A**) Relative mRNA expression of epidermal stem-cell markers in tail epidermis from control and $Hdac1^{\Delta/\Delta ep}$ $Hdac2^{\Delta/+ep}$ littermates. $n = 3–4$. $P = 0.0054$, 0.0006, 0.0022, 0.0081 and 0.0008. (**B**) IF co-staining of label-retaining (BrdU + ) bulge cells (CD34) in HF (left panels) and IFE (right panels) of control and $Hdac1^{\Delta/\Delta ep}$ $Hdac2^{\Delta/+ep}$ mice (P34) after 72 h BrdU pulse (P6) and 25 days chase. Nuclei were counterstained with DAPI. Scale bar: 100 μm. Quantification of BrdU is shown on the right of each staining picture. $n = 4$. $P = 0.0241$, 0.0030 and 0.0646. (**C**) Back skin sections of control and $Hdac1^{\Delta/\Delta ep}$ $Hdac2^{\Delta/+ep}$ mice (P30) labelled with an antibody against c-Myc. Nuclei were counterstained with DAPI. Scale bar: 50 μm. (**D**) Expression of c-Myc in epidermal tail nuclear cell extracts of control, $Hdac1^{\Delta/+ep}$ $Hdac2^{\Delta/\Delta ep}$ and $Hdac1^{\Delta/\Delta ep}$ $Hdac2^{\Delta/+ep}$ mice was analysed on immunoblots. Lamin B served as a loading control. Immunoblot signals of control and $Hdac1^{\Delta/\Delta ep}$ $Hdac2^{\Delta/+ep}$ mice were quantified by densitometric scanning are shown relative to the Lamin B signal on the right of the panel. $n = 3$. (**A**, **B**, **D**) Error bars indicate s.d. Source data for this figure is available on the online supplementary information page.

contrary, *Lgr6* an SC marker that is postnatally contributing to SG and IFE growth (Snippert *et al*, 2010; Schuijers and Clevers, 2012) showed increased expression in the mutant epidermis (Figure 4A). To examine whether there was a change in SC fate in $Hdac1^{\Delta/\Delta ep}$ $Hdac2^{\Delta/+ep}$ epidermis, we performed BrdU pulse-chase experiments to analyse slow cycling label retaining cells (LRCs) and their progeny. Control and $Hdac1^{\Delta/\Delta ep}$ $Hdac2^{\Delta/+ep}$ mice (P6) were pulsed for 72 h with BrdU, followed by a 25-day chase. Proliferating cells lose the BrdU label whereas slow cycling LRCs retain the BrdU label. To identify SCs in the bulge, co-stainings for BrdU and CD34 were performed. As shown in Figure 4B, we detected a reduction in BrdU$^+$/CD34$^+$ cells in the HFs of $Hdac1^{\Delta/\Delta ep}$ $Hdac2^{\Delta/+ep}$ mice, which is at least in part caused by increased cell death (Figure 2D). At the same time, we

observed an increase in BrdU$^+$ cells in the IFE and (albeit not significant) in SGs of mutant mice (Figure 4B). These findings reveal profound changes in epidermal lineages inflicted by enhanced apoptosis in the HFs, increased proliferation in the IFE and/or mobilization and redirection of epidermal SCs in $Hdac1^{\Delta/\Delta ep}$ $Hdac2^{\Delta/+ep}$ mice. The increased proliferation in the IFE suggests that the bulge SCs are able to contribute to IFE regeneration. This prompted us to perform wound-healing assays by introduction of full-thickness excisional wounds in the back skin. Notably, mutant mice display a delay in wound healing and developed keratoacanthoma-like lesions with massive keratin deposition around the wound margins (Supplementary Figures S5B and C).

The phenotype of $Hdac1^{\Delta/\Delta ep}$ $Hdac2^{\Delta/+ep}$ mice with hyperproliferation of the IFE, increased SG formation

and impaired hair development is reminiscent of c-Myc overexpressing mice, which display epidermal hyperproliferation and differentiation along the SG and IFE lineages at the expense of HF development (Arnold and Watt, 2001; Waikel *et al*, 2001). Therefore, we analysed c-Myc expression in the epidermis of control and mutant mice. $Hdac1^{\Delta/\Delta ep}$ $Hdac2^{\Delta/+ep}$ mice displayed increased c-Myc protein levels in the epidermis detectable both by IHC and by immunoblot analysis, whereas $Hdac1^{\Delta/+ep}$ $Hdac2^{\Delta/\Delta ep}$ and control epidermis showed no c-Myc induction (Figures 4C and D). Accordingly, the c-Myc protein was also strongly expressed in spontaneously formed tumours of older mutant animals (Figure 1D; Supplementary Figure S5D). Collectively, our results obtained with $Hdac1^{\Delta/\Delta ep}$ $Hdac2^{\Delta/+ep}$ mice are comparable with data published for c-Myc overexpressing mice which display increased proliferation and differentiation along the epidermal and sebaceous lineages at the expense of HF differentiation (Arnold and Watt, 2001; Waikel *et al*, 2001).

### Deregulated gene expression in $Hdac1^{\Delta/\Delta ep}$ $Hdac2^{\Delta/+ep}$ epidermis

Given the different epidermal phenotypes in mice with *Hdac1* disruption in the presence and absence of up-regulated HDAC2 levels, we compared gene expression profiles of the epidermis of $Hdac1^{\Delta/\Delta ep}$, $Hdac1^{\Delta/\Delta ep}$ $Hdac2^{\Delta/+ep}$ mice and the corresponding control littermates. A global expression analysis showed that in $Hdac1^{\Delta/\Delta ep}$ mice only 79 genes were deregulated compared to control littermates ($P < 0.05$, at least two-fold change in expression) (Figure 5A; Supplementary Table S2). In contrast, 3749 genes were deregulated in the epidermis of $Hdac1^{\Delta/\Delta ep}$ $Hdac2^{\Delta/+ep}$ mice suggesting that up-regulated HDAC2 can compensate for most of the lost regulatory function of HDAC1 in $Hdac1^{\Delta/\Delta ep}$ epidermis (Figure 5B; Supplementary Table S2). We categorized the up-regulated genes by functional gene ontology (GO) analysis as shown in Supplementary Table S3. In agreement with the observed phenotype, genes in categories such as cell proliferation and keratinization were enriched in the group of up-regulated genes. In a recent publication, Nascimento *et al* (2011) reported a functional link between Sin3A and c-Myc in the regulation of epidermal development. In the absence of Sin3A in the epidermis, c-Myc was up-regulated resulting in a phenotype that is reminiscent of $Hdac1^{\Delta/\Delta ep}$ $Hdac2^{\Delta/+ep}$ mice. Consistent with this idea we found that more than one third of the genes deregulated in $Sin3a^{\Delta/\Delta ep}$ epidermis (Nascimento *et al*, 2011) were also deregulated in the $Hdac1^{\Delta/\Delta ep}$ $Hdac2^{\Delta/+ep}$ epidermis (Figure 5C). Furthermore, a subset of these genes (166) have previously been shown to be deregulated upon overexpression of c-Myc in the epidermis and a significant number of genes are commonly deregulated in $Hdac1^{\Delta/\Delta ep}$ $Hdac2^{\Delta/+ep}$ and c-Myc overexpressing epidermis (Figure 5C) (Frye *et al*, 2003). These data suggest that an impaired Sin3A co-repressor function contributes to the phenotype of $Hdac1^{\Delta/\Delta ep}$ $Hdac2^{\Delta/+ep}$ mice.

### Partially impaired co-repressor function in $Hdac1^{\Delta/\Delta ep}$ $Hdac2^{\Delta/+ep}$ epidermis

Therefore, we next investigated by co-immunoprecipitation experiments whether HDAC1/HDAC2 co-repressor activity was affected in $Hdac1^{\Delta/\Delta ep}$ $Hdac2^{\Delta/+ep}$ keratinocytes. Remarkably, deacetylase activities associated with Sin3A,

the NuRD complex component MTA2 and CoREST were all reduced in $Hdac1^{\Delta/\Delta ep}$ $Hdac2^{\Delta/+ep}$ keratinocytes (Figure 5D). Immunoblot analysis showed reduced protein levels of Sin3A and MTA2 in $Hdac1^{\Delta/\Delta ep}$ $Hdac2^{\Delta/+ep}$ keratinocytes, whereas CoREST expression was not affected (Figures 5E and H). The reduction in MTA2 and Sin3A protein levels was not caused by reduced mRNA expression but by lower protein stability as shown in cycloheximide protein stability assays (Supplementary Figures S6A and B). In contrast, $Hdac1^{\Delta/+ep}$ $Hdac2^{\Delta/\Delta ep}$ keratinocytes showed no significant changes in co-repressor associated deacetylase activities (Figures 5F, G and I). Thus, loss of HDAC1 in the absence of elevated HDAC2 leads to strongly reduced deacetylase activity and partial destabilization of HDAC1/HDAC2 repressor complexes.

Sin3A (and associated proteins) has been previously shown to cause deacetylation and destabilization of the c-Myc protein (Nascimento *et al*, 2011). Of note, the up-regulation of c-Myc protein in the $Hdac1^{\Delta/\Delta ep}$ $Hdac2^{\Delta/+ep}$ epidermis was not accompanied by increased *c-Myc* mRNA levels suggesting a post-transcriptional regulatory mechanism (Supplementary Figure S6C). In accordance with this hypothesis, HDAC inhibition with the class I-specific deacetylase inhibitor MS-275, which has a preference for HDAC1 (Bertrand, 2010) led to increased c-Myc protein levels and protein stability without affecting *c-Myc* mRNA expression (Supplementary Figure S6D). These data, together with the findings of Nascimento *et al* (2011), indicate a crucial function of the Sin3A co-repressor complex and associated HDAC1/(HDAC2) activity in preventing inappropriately high levels of c-Myc protein in the epidermis.

Several members of the differentiation-associated members of the Sprr and LCE protein families which had been found to be negatively regulated by Sin3A are also up-regulated in the $Hdac1^{\Delta/\Delta ep}$ $Hdac2^{\Delta/+ep}$ epidermis (Nascimento *et al*, 2011) (Supplementary Table S1). To ask how the impaired Sin3A co-repressor function affects the regulation of genes commonly deregulated in Sin3A-deficient and $Hdac1^{\Delta/\Delta ep}$ $Hdac2^{\Delta/+ep}$ epidermis, we performed chromatin immunoprecipitation (ChIP) assays. The association of HDAC1, HDAC2 and Sin3A and the presence of histone H3K9ac and H4ac marks was analysed for up-regulated genes representing the EDC (*Sprr2h*, *S100a8*, *S100a9*, *Lce3b*), SG (*Klk6*), epidermal proliferation markers (*Epgn*, *Ada*), and the non-regulated control *Vstm2l* with chromatin isolated from epidermis of control, $Hdac1^{\Delta/\Delta ep}$ and $Hdac1^{\Delta/\Delta ep}$ $Hdac2^{\Delta/+ep}$ mice. In wild-type controls, HDAC1, HDAC2 and Sin3A were present at the promoter regions of *Sprr2h*, *S100a8*, *S100a9*, *Lce3b*, *Klk6*, *Epgn*, *Ada*, *Hoxc13* and *Msx2* (Figure 6; Supplementary Figures S7A and B). In $Hdac1^{\Delta/\Delta ep}$ epidermis, where these genes are not deregulated, increased amounts of HDAC2 were associated with these promoters in the absence of HDAC1. In contrast, in $Hdac1^{\Delta/\Delta ep}$ $Hdac2^{\Delta/+ep}$ epidermis the presence of Sin3A and HDAC2 at the *Sprr2h* and *Epgn* promoters was reduced concomitant with an increase in local histone acetylation and gene expression. On the other hand, the down-regulated hair development regulator genes (*Hoxc13*, *Msx2*) showed reduced histone acetylation levels upon loss of HDAC1 in $Hdac1^{\Delta/\Delta ep}$ $Hdac2^{\Delta/+ep}$ epidermis suggesting a potential positive role of the deacetylase for the expression of these genes (Supplementary Figure S7B). We conclude that increased presence of HDAC2 at these target genes can

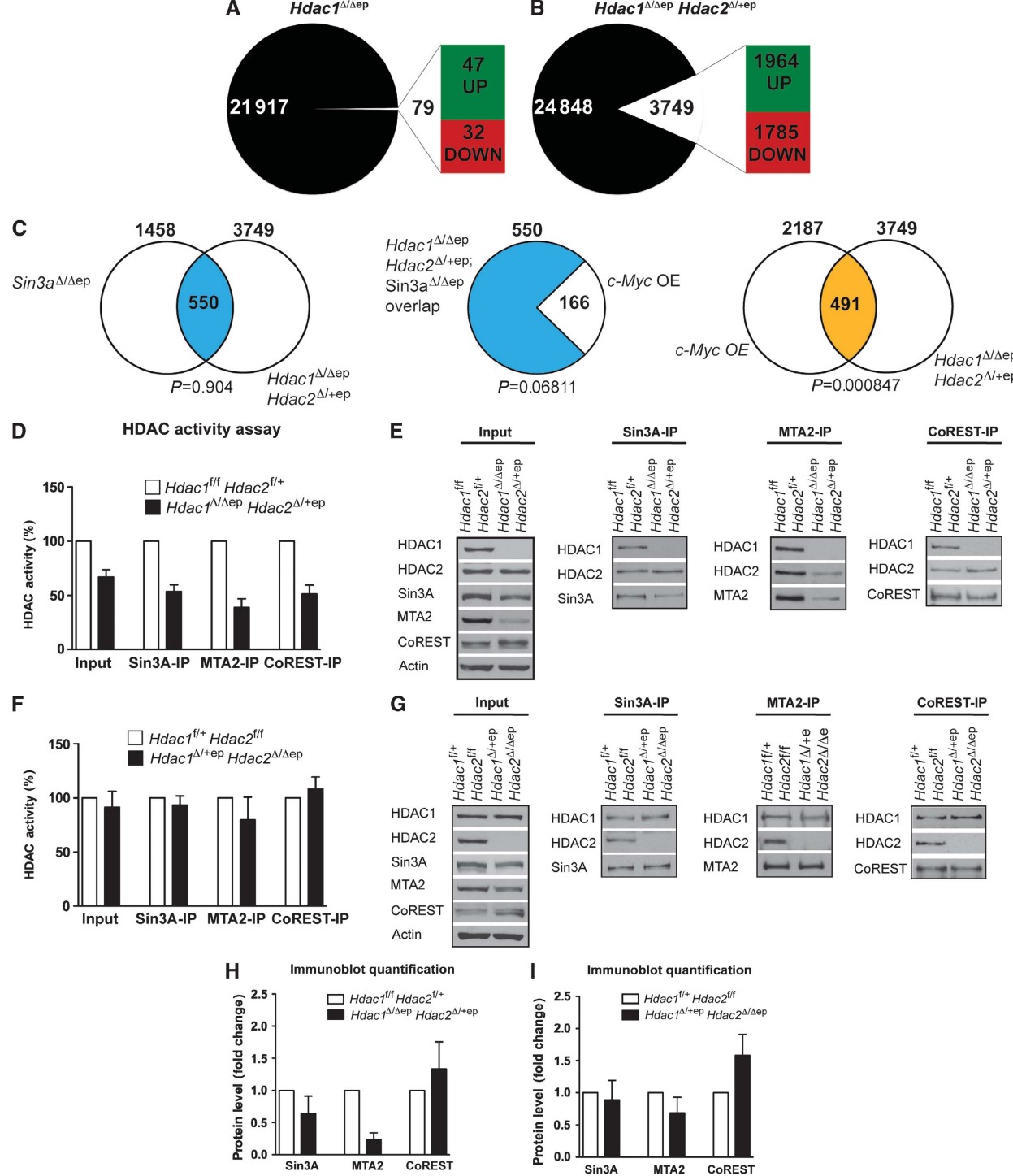

**Figure 5** Gene expression and changes in repressor complex function in $Hdac1^{\Delta/\Delta ep}\ Hdac2^{\Delta/+ep}$ mice. Agilent microarray gene expression analysis of $Hdac1^{\Delta/\Delta ep}$ ($n=4$) (**A**), $Hdac1^{\Delta/\Delta ep}\ Hdac2^{\Delta/+ep}$ ($n=3$) (**B**) and the corresponding control mice. RNA was isolated from adult tail epidermis from three animals of each genotype. In all, 79 annotated genes in $Hdac1^{\Delta/\Delta ep}$ mice (**A**) and 3749 genes in $Hdac1^{\Delta/\Delta ep}\ Hdac2^{\Delta/+ep}$ mice (**B**) were >2-fold deregulated. (**C**) Overlap of deregulated genes of $Hdac1^{\Delta/\Delta ep}\ Hdac2^{\Delta/+ep}$ mice with deregulated genes of $Sin3a^{\Delta/\Delta ep}$ mice (Nascimento *et al*, 2011). Out of this overlap of 550 genes, 166 genes were deregulated in c-Myc overexpressing epidermis (Frye *et al*, 2003). Overlap of deregulated genes in c-Myc overexpressing (OE) epidermis with deregulated genes of $Hdac1^{\Delta/\Delta ep}\ Hdac2^{\Delta/+ep}$ mice. Significance of overlaps was determined by a hypergeometric probability test. (**D–G**) Sin3A, NuRD and CoREST complexes were immunoprecipitated from keratinocyte extracts of control, $Hdac1^{\Delta/\Delta ep}\ Hdac2^{\Delta/+ep}$ (**D, E**) and $Hdac1^{\Delta/+ep}\ Hdac2^{\Delta/\Delta ep}$ (**F, G**) mice with antibodies against Sin3A, MTA2 and CoREST. (**D, F**) Complex-associated HDAC activity was measured in input and IPs. Controls are set to 100%. $n=4$. (**E, G**) Immunoprecipitated proteins were analysed on immunoblots with HDAC1, HDAC2, Sin3A, MTA2 and CoREST antibodies. β-actin served as a loading control. (**H, I**) Immunoblot signals for Sin3A, MTA2 and CoREST in $Hdac1^{\Delta/\Delta ep}\ Hdac2^{\Delta/+ep}$ (**H**), $Hdac1^{\Delta/+ep}\ Hdac2^{\Delta/\Delta ep}$ (**I**) and the corresponding control keratinocytes were quantified by densitometric scanning and are shown relative to the β-actin signal. The values of wild-type controls were set to 1 ($n=4$). (**D, F, H**) Data are mean ± s.d. Source data for this figure is available on the online supplementary information page.

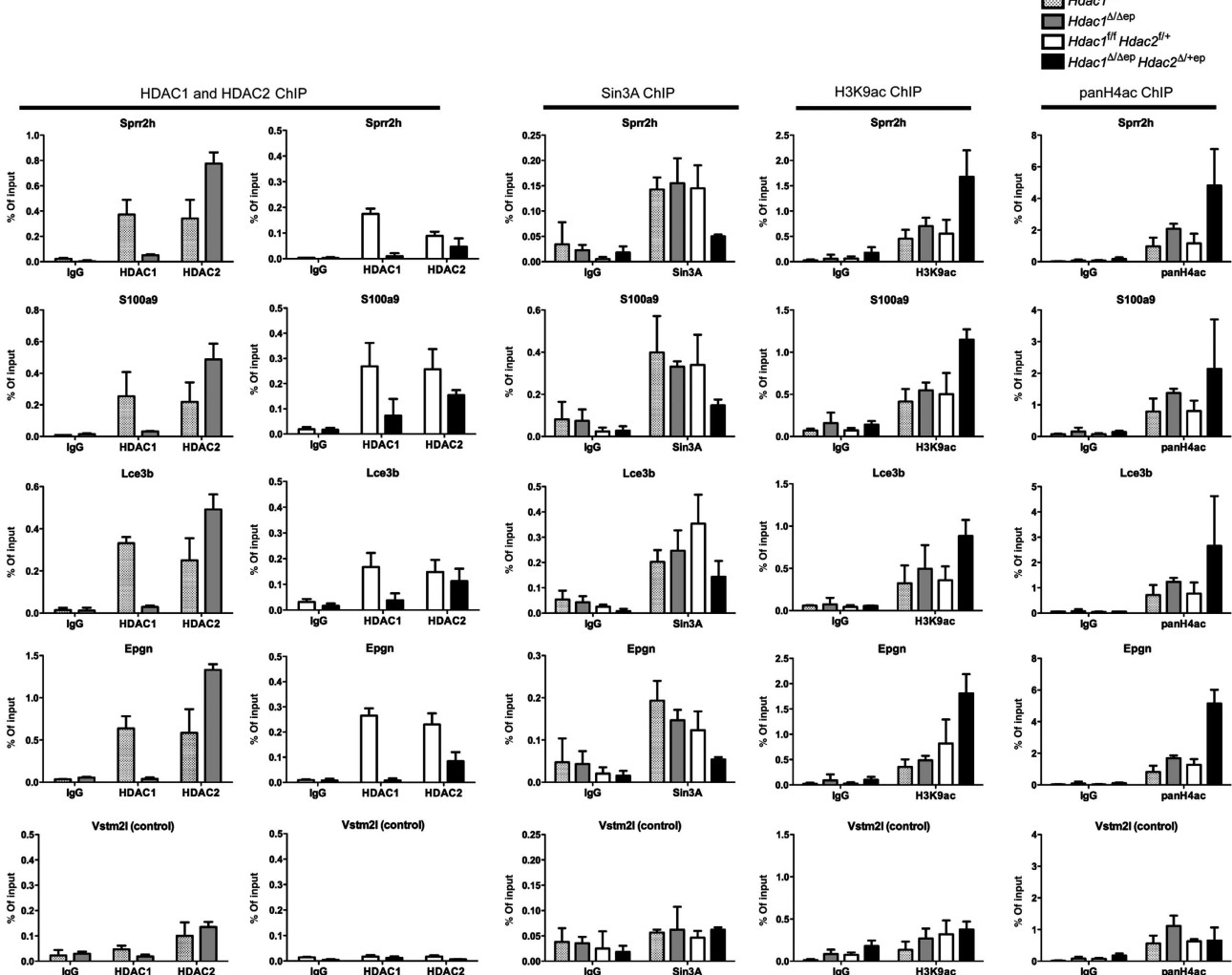

**Figure 6** ChIP analysis of Sin3A/HDAC1/HDAC2 target genes. Chromatin from littermate controls, $Hdac1^{\Delta/\Delta ep}$ and $Hdac1^{\Delta/\Delta ep}\ Hdac2^{\Delta/+ep}$ epidermis was immunoprecipitated with antibodies specific for HDAC1, HDAC2, Sin3A, histone H3K9ac, pan-histone H4ac or IgG as a control. Precipitated DNA was analysed by qRT-PCR with primers specific for Sprr2h, S100a9, Lce3b, Epgn and Vstm2l (control) promoter regions. Data are representative for at least two independent experiments. $n = 2$–$5$. Mean value with s.d. is shown.

compensate for the loss of HDAC1 in $Hdac1^{\Delta/\Delta ep}$ mice, whereas in $Hdac1^{\Delta/\Delta ep}\ Hdac2^{\Delta/+ep}$ mice lower Sin3A and HDAC2 levels fail to fully exert the Sin3A/HDAC1/HDAC2 co-repressor function.

### Enhanced tumour development in K5-SOS Hdac1$^{\Delta/\Delta ep}$ mice

HDAC1 and HDAC2 have been shown to be crucial for cell proliferation and are therefore considered as promising targets for anti-tumour therapy (Lagger *et al*, 2002; Haberland *et al*, 2010; Yamaguchi *et al*, 2010). However, the data shown above point towards an anti-proliferative function of HDAC1 during epidermal development and homeostasis. We therefore examined the effect of ablation of HDAC1 and HDAC2 in the genetic K5-SOS skin tumour model (Sibilia *et al*, 2000). In this system, the ras pathway is constitutively activated in the epidermis due to a dominant form of Son of Sevenless (SOS) expressed under the control of the *K5* promoter (*K5-SOS*). In the presence of a functional epidermis growth factor receptor (EGFR), required to provide an essential survival signal to tumour cells, *K5-SOS* mice develop skin tumours that share features of

human squamous cell carcinomas (Sibilia *et al*, 2000). Papillomas develop predominantly on the tail, the paw, behind the ears, and at sites subjected to scratching and biting. Notably, *K5-SOS Hdac1*$^{\Delta/\Delta ep}$ mice showed accelerated onset of tumour development and a significant increase in relative tail tumour weight (Figures 7A–C). In accordance with our previous observations, deletion of HDAC1 led to an up-regulation of HDAC2 in *K5-SOS Hdac1*$^{\Delta/\Delta ep}$ epidermis (Figure 7D). HDAC activity assays revealed a significant reduction in total cellular deacetylase activity in the epidermis of the *K5-SOS Hdac1*$^{\Delta/\Delta ep}$ mice (Figure 7E). HDAC1-deficient tumours showed hyperproliferation of the basal and suprabasal layers as determined by Ki67 staining and reduced expression of the epidermal differentiation marker K10 (Figures 7F and G). Importantly, co-repressor associated deacetylase was significantly reduced for Sin3A, NuRD and Co-REST complexes in HDAC1-deficient tumours compared to wild-type tumours (Figures 7H and I). In accordance with the link between Sin3A function and c-Myc expression, we detected increased levels of c-Myc protein and the c-Myc target Skp2 in *K5-SOS Hdac1*$^{\Delta/\Delta ep}$ tumours (Figures 7J and K).

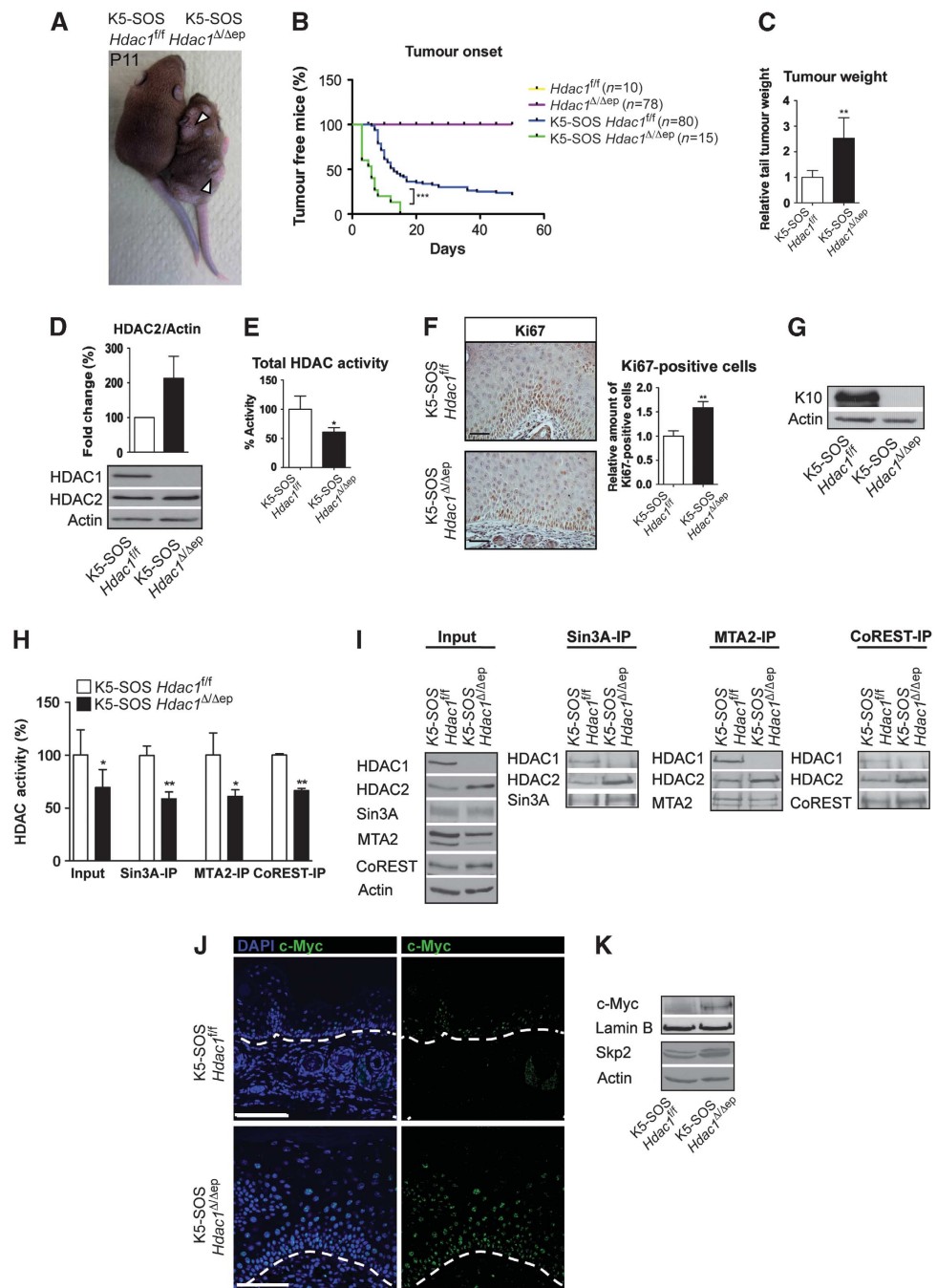

**Figure 7** Increased tumour development in *K5-SOS Hdac1*$^{\Delta/\Delta ep}$ mice. (**A**) Picture of littermate control and *K5-SOS Hdac1*$^{\Delta/\Delta ep}$ mice at P11. Arrows indicate sites of premature tumour appearance on mutant mice. (**B**) Tumour appearance curves of control and *K5-SOS Hdac1*$^{\Delta/\Delta ep}$ mice ($P = 0.0001$) and (**C**) their relative tail tumour weight at P11. $n = 3–5$. $P = 0.0064$. (**D**) Immunoblot analysis of HDAC1 and HDAC2 protein expression, with β-actin as a loading control and quantification of the immunoblot signal. $n = 3$. (**E**) HDAC activity measured in SOS tail epidermis of *K5-SOS* control and *K5-SOS Hdac1*$^{\Delta/\Delta ep}$ mice. $n = 4$, $P = 0.0128$. (**F**) IHC staining and quantification of Ki67 in tail skin of control and *K5-SOS Hdac1*$^{\Delta/\Delta ep}$ (P11). Scale bar: 50 μm. $n = 3$, $P = 0.0035$. (**G**) Immunoblot analysis of K10 expression in epidermal extracts from control and *K5-SOS Hdac1*$^{\Delta/\Delta ep}$ mice. β-actin was used as a loading control. (**H, I**) Analysis of co-repressor complexes and associated deacetylase activity in control and *K5-SOS Hdac1*$^{\Delta/\Delta ep}$ mice. Input and immunoprecipitated proteins were analysed on immunoblots with HDAC1, HDAC2, Sin3A, MTA2 and CoREST antibodies. β-Actin served as a loading control (**H**: $n = 3$, $P = 0.0103$, 0.003, 0.0358 and 0.0029). (**J**) Skin tumour sections of control and *K5-SOS Hdac1*$^{\Delta/\Delta ep}$ mice (P11) were labelled with c-Myc antibody and nuclei were counterstained with DAPI. Scale bar: 100 μm. (**K**) Expression of c-Myc and Skp2 in nuclear and whole-cell extracts respectively of control and *K5-SOS Hdac1*$^{\Delta/\Delta ep}$ mice epidermal tumours. Lamin B and β-actin served as a loading control. An at least three-fold up-regulation of c-Myc was observed in three independent experiments. (**C–F,H**) Data represent means ± s.d. Source data for this figure is available on the online supplementary information page.

In contrast, ablation of HDAC2 in the epidermis of *K5-SOS* mice had no effect on tumour appearance and tumour weight (Figures 8A–C). While HDAC1 protein levels were up-regulated in the absence of HDAC2, total cellular and co-repressor associated HDAC activities were unchanged in *K5-SOS Hdac2*$^{\Delta/\Delta ep}$ epidermis (Figures 8D, E, H and I).

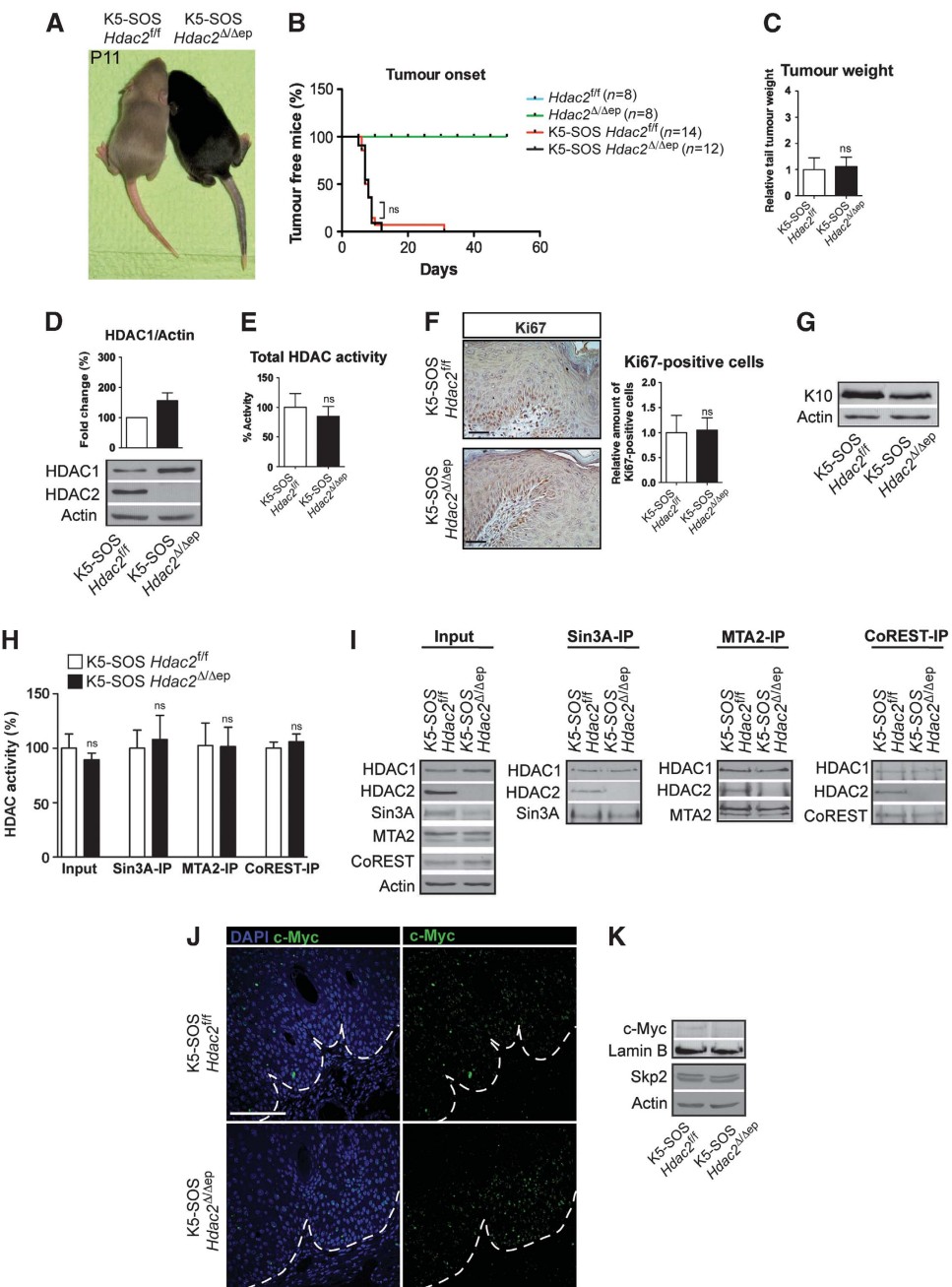

**Figure 8** Loss of HDAC2 in the epidermis has no effect on tumour development in *K5-SOS* mice. (**A**) Picture of littermate control and *K5-SOS Hdac2*$^{\Delta/\Delta ep}$ mice at P11. (**B**) Tumour appearance curves of control and *K5-SOS Hdac2*$^{\Delta/\Delta ep}$ mice. Note that in the genetic background of *Hdac2*$^{f/f}$ mice tumours appear earlier compared to background of *Hdac1*$^{f/f}$ mice. $P = 0.8829$. (**C**) Relative tail tumour weight of control and *K5-SOS Hdac2*$^{\Delta/\Delta ep}$ mice at P11. $n = 3$–$5$, $P = 0.7293$. (**D**) Immunoblot analysis of HDAC1 and HDAC2 protein expression, with β-actin as a loading control and the quantified immunoblot signal. $n = 3$. (**E**) HDAC activity measured in SOS tail epidermis of *K5-SOS* control and *K5-SOS Hdac2*$^{\Delta/\Delta ep}$ mice. $n = 3$, $P = 0.3996$. (**F**) IHC staining and quantification of Ki67 in tail skin of control and *K5-SOS Hdac2*$^{\Delta/\Delta ep}$ (P11). Scale bar: 50 μm. $n = 3$, $P = 0.8483$. (**G**) Immunoblot analysis of K10 expression in protein extracts from control and *K5-SOS Hdac2*$^{\Delta/\Delta ep}$ epidermis. β-actin was used as a loading control. (**H, I**) Analysis of co-repressor complexes and associated deacetylase activity in control and *K5-SOS Hdac2*$^{\Delta/\Delta ep}$ mice. Input and immunoprecipitated proteins were analysed on immunoblots with HDAC1, HDAC2, Sin3A, MTA2 and CoREST antibodies. (H: $n = 3$, β-actin served as a loading control. $P = 0.2697$, $0.6416$, $0.9631$ and $0.4477$). (**J**) Skin tumour sections of control and *K5-SOS Hdac2*$^{\Delta/\Delta ep}$ mice (P11) were labelled with c-Myc antibody and nuclei were counterstained with DAPI. Scale bar: 100 μm. (**K**) Expression of c-Myc and Skp2 in nuclear or whole-cell epidermal extracts respectively of control and *K5-SOS Hdac2*$^{\Delta/\Delta ep}$ mice tumours. Lamin B and β-actin served as a loading control. (**C–F, H**) Mean value with s.d. is depicted. Source data for this figure is available on the online supplementary information page.

In agreement with the unaffected tumour size, expression of c-Myc, Skp2 and the proliferation marker Ki67 was comparable to littermate controls whereas the differentiation marker K10 was slightly reduced in *K5-SOS Hdac2*$^{\Delta/\Delta ep}$ tumours (Figures 8F, G, J and K). Thus, HDAC1 but not HDAC2 negatively controls tumour cell proliferation in the

epidermis. These results point towards a novel and unexpected role of HDAC1 as a tumour suppressor in the epidermis.

## Discussion

### Individual and redundant functions of HDAC1 and HDAC2 in the epidermis

In this report, we analysed individual and redundant functions of HDAC1 and HDAC2 in epidermal development, homeostasis and tumorigenesis. Strikingly, *K5-Cre* mediated deletion of three of the four *Hdac1/Hdac2* alleles had different consequences depending on the remaining allele. Mice with a single *Hdac1* allele in the epidermis ($Hdac1^{\Delta/+ep}\ Hdac2^{\Delta/\Delta ep}$) showed no significant changes in HDAC activity, proliferation and differentiation in the epidermis, indicating that one *Hdac1* allele is sufficient to maintain proper epidermal development. In contrast, mice with a single *Hdac2* allele ($Hdac1^{\Delta/\Delta ep}\ Hdac2^{\Delta/+ep}$) displayed a severe developmental phenotype accompanied by a significant reduction in total HDAC activity. These data reveal a predominant role of HDAC1 during epidermal development. The severe developmental phenotype of $Hdac1^{\Delta/\Delta ep}\ Hdac2^{\Delta/+ep}$ mice using *K5-Cre* was also described in a recent report using *K14-Cre* albeit without description of a molecular mechanism (Hughes *et al*, 2013) but were not uncovered in another report using the same Cre transgene (LeBoeuf *et al*, 2011). The reasons for this discrepancy are unclear.

HDAC1 was identified as the major deacetylase during embryonic development (Lagger *et al*, 2002), in B cells (Reichert *et al*, 2012) and T cells (Grausenburger *et al*, 2010; Dovey *et al*, 2013; Heideman *et al*, 2013). On the contrary, a single *Hdac2* but not *Hdac1* allele has been shown to be sufficient for normal oocyte (Ma *et al*, 2012) and brain development (Hagelkruys, Lagger *et al*, manuscript in revision). Taken together, these findings indicate that HDAC1 and HDAC2 do not only have redundant but also specific functions in differentiation and development.

Loss of HDAC1 in the absence of compensating increased levels of HDAC2 in $Hdac1^{\Delta/\Delta ep}\ Hdac2^{\Delta/+ep}$ epidermis led to a severe reduction in HDAC activity. This was in part due to destabilization of certain components of the Sin3 and NuRD co-repressor complexes such as Sin3A, MTA1 and MTA2. Thus, in addition to their catalytic functions HDAC1 and HDAC2 seem to also have a scaffolding function required for the stability of epidermal co-repressor complexes. Notably, a similar dosage-dependent effect of HDAC1/HDAC2 ablation on HDAC activity and repressor complex function was recently observed in mouse T cells (Dovey *et al*, 2013). HDAC1-deficient T cells with only one *Hdac2* allele exhibited a significant reduction in HDAC activity associated with Sin3A and NuRD complexes and reduced Sin3A and MTA2 protein levels. These findings support the hypothesis that beyond their enzymatic function HDAC1/2 are also crucial for the structural integrity of repressor complexes.

### Control of epidermal proliferation, differentiation and lineage faith by HDAC1/2 co-repressor complexes

$Hdac1^{\Delta/\Delta ep}Hdac2^{\Delta/+ep}$ mice exhibited profound changes in the development of all three epidermal lineages resulting in epidermal hyperproliferation and hyperkeratosis, accompanied by hair loss and SG enlargement. The most remarkable phenotype of mutant mice is the dramatic alopecia.

$Hdac1^{\Delta/\Delta ep}\ Hdac2^{\Delta/+ep}$ HFs are generated with similar frequency but fail to properly enter and transit the first hair cycle. Already during morphogenesis mutant HFs show increased levels of p53 and apoptosis resulting in HF atrophy. Accordingly, the absence of HDAC1/HDAC2 in the epidermis was shown to lead to high levels of acetylated p53 (LeBoeuf *et al*, 2011) suggesting a direct or indirect role of the two class I deacetylases as negative regulators of p53 activity. Interestingly, in contrast to mutant HFs the IFE of $Hdac1^{\Delta/\Delta ep}$ $Hdac2^{\Delta/+ep}$ mice is more resistant to p53 induced apoptosis. This might be one explanation for the lineage-specific differences in the $Hdac1^{\Delta/\Delta ep}Hdac2^{\Delta/+ep}$ epidermis.

On the other hand, up-regulation of c-Myc, a central regulator of epidermal lineage commitment, contributes to impaired epidermal homeostasis of $Hdac1^{\Delta/\Delta ep}\ Hdac2^{\Delta/+ep}$ mice. Overexpression of c-Myc was shown to mobilize SCs from the SC compartment and to promote proliferation as well as differentiation of the IFE and SG lineage at the expense of HF development (Arnold and Watt, 2001; Lo Celso *et al*, 2008; Berta *et al*, 2010). Accordingly, we observed SG hypertrophy and increased levels of the SC marker Lgr6 in mutant mice. We cannot fully exclude that wounding contributes to the mobilization of HF SCs to the IFE of $Hdac1^{\Delta/\Delta ep}\ Hdac2^{\Delta/+ep}$ mice; however, we could not detect increased numbers of infiltrating immune cells in the mutant epidermis. $Hdac1^{\Delta/\Delta ep}Hdac2^{\Delta/+ep}$ mice phenocopy to a large extent mice with epidermal deletion of Sin3A (Nascimento *et al*, 2011). We propose that the partially impaired function of the Sin3A co-repressor complex in the $Hdac1^{\Delta/\Delta ep}Hdac2^{\Delta/+ep}$ epidermis has two consequences.

First, this leads to stabilization and increased expression of the c-Myc protein. Sin3A was shown to counteract c-Myc activity by causing the deacetylation of c-Myc protein (Nascimento *et al*, 2011), while acetylation has been shown to enhance the stability of the c-Myc protein (Vervoorts *et al*, 2003; Patel *et al*, 2004). In agreement with the idea that the Sin3A/HDAC1/HDAC2 co-repressor complex attenuates c-Myc activity by reducing protein stability, we found that the c-Myc protein can be stabilized by HDAC inhibition and was increased in the $Hdac1^{\Delta/\Delta ep}Hdac2^{\Delta/+ep}$ epidermis without changes in mRNA levels. In line with an important role for c-Myc as a contributor to the impaired epidermal development, we discovered a significant overlap between genes deregulated in $Hdac1^{\Delta/\Delta ep}Hdac2^{\Delta/+ep}$ epidermis with genes deregulated in c-Myc overexpressing mice (Frye *et al*, 2003).

Second, the reduced amount of Sin3A and the impaired Sin3A co-repressor function led to derepression of differentiation specific genes, thereby contributing to the hyperkeratosis of the mutant epidermis. ChIP experiments show that reduced Sin3A recruitment and uncompensated loss of HDAC1 at EDC genes resulted in increased local histone acetylation and gene expression. Given the partially impaired function of CoREST and NuRD complexes and the large number of deregulated genes other HDAC1/HDAC2 recruiting factors and complexes in addition to the Sin3A complex might also play important roles in epidermal development and homeostasis.

### HDAC1 acts as a tumour suppressor in the epidermis

In an SOS-driven skin tumour model deletion of HDAC1 in the epidermis led to enhanced tumorigenesis associated with

increased proliferation and reduced differentiation, whereas ablation of HDAC2 had no effect. $Hdac1^{\Delta/\Delta ep}$ mice did not display an obvious developmental phenotype but the frequently observed scar formation at the tail and the accelerated tumour development in the K5-SOS tumour model suggest that under mechanical or oncogenic stress conditions HDAC2 cannot fully compensate for the loss of HDAC1 in the epidermis. In accordance with this idea, total deacetylase and co-repressor activity was unchanged in $Hdac1^{\Delta/\Delta ep}$ epidermis but clearly reduced in HDAC1-deficient tumours. In particular, the impaired activity of the Sin3A co-repressor complex and the concomitant overexpression of the c-Myc in HDAC1- but not HDAC2-deficient papillomas provide strong evidence for crucial function for Sin3A and HDAC1 as a negative regulator of the proto-oncogene c-Myc. Thus, the hyperproliferation and spontaneous tumour development observed in $Hdac1^{\Delta/\Delta ep}Hdac2^{\Delta/+ep}$ epidermis and in $Hdac1$-deficient SOS skin tumours indicates a novel role of HDAC1 as a tumour suppressor in the skin. These observations show parallels with two recent studies on the function of HDAC1/HDAC2 in T cells. Deletion of two $Hdac1$ alleles and a single $Hdac2$ allele by $Lck$-$Cre$ resulted in neoplastic transformation of immature T cells (Dovey *et al*, 2013; Heideman *et al*, 2013). In this study, enhanced c-Myc expression was detectable at the mRNA level and mostly due to trisomy of chromosome 15. The authors concluded that HDAC1 and HDAC2 control genomic stability (Dovey *et al*, 2013; Heideman *et al*, 2013) and p53 activity controlling functions (Heideman *et al*, 2013) in a dosage-dependent manner.

The function of HDAC1 as a negative regulator of proliferation in the epidermis is in contrast to some earlier studies in which HDAC1 deficiency in ES cells or fibroblasts showed reduced rates of proliferation due to increased levels of the CDK inhibitor p21 (Lagger *et al*, 2002; Zupkovitz *et al*, 2010). Similarly, loss of HDAC1 in human tumour cells led to reduced proliferation and elevated apoptosis (Glaser *et al*, 2003; Senese *et al*, 2007). On the contrary, it was shown that the conditional loss of HDAC1 in T cells results in increased proliferation (Grausenburger *et al*, 2010) and HDAC1-deficient murine teratomas display enhanced proliferation and reduced differentiation (Lagger *et al*, 2010).

In summary, we have discovered a crucial function of HDAC1 in epidermal homeostasis and as a tumour suppressor in skin cancer. Our results have potential consequences for medical applications and the use of HDAC inhibitors in tumour therapy. For instance, low expression of HDAC1/ HDAC2 in the epidermis might cause a predisposition towards neoplasia. Furthermore, HDAC inhibitors that mostly target HDAC1 could induce undesired effects such as hyperproliferation. Finally, drugs that target both HDAC1 and HDAC2 might be promising since loss of both enzymes invariably induces cell death in all proliferating cell types and tissues tested so far.

## Materials and methods

### Animal care and transgenic mouse lines
All mouse lines were bred to a mixed genetic background of C57BL/ 6J × 129SV. Keratin 5 Cre (K5-Cre) mice (Ramirez *et al*, 2004) were mated to $Hdac1^{f/f}$ and $Hdac2^{f/f}$ mice (Yamaguchi *et al*, 2010) to generate mice with deletions of $Hdac1$ and/or $Hdac2$ alleles in the epidermis. K5-SOS tumour mice with hypomorphic EGFR alleles (EGFR$^{wa2/wa2}$) (Sibilia *et al*, 2000) were crossed to $Hdac1^{f/f}$ and

$Hdac2^{f/f}$ mice. Mice were kept in the animal facilities of the MFPL and the Medical University of Vienna in accordance with institutional policies and federal guidelines.

### Epidermis isolation and keratinocyte culture
Murine tail skin was isolated and incubated dermis side down in PBS containing 5 mg/ml Dispase (Roche) for 2 h at 4°C. Epidermis was removed from dermis, washed in PBS and snap frozen. Tissue was further used for protein or RNA isolation. Mouse epidermal keratinocytes were isolated as previously described (Sibilia *et al*, 2000) and cultured in keratinocyte growth medium (KGM BulletKit©, Clonetics©) containing 8% chelated fetal calf serum and 0.05 mM CaCl$_2$ on collagen-fibronectin coated dishes.

### RNA isolation and real-time PCR analysis
Total RNA was isolated following the manufacturer's instructions (TRIzol, Invitrogen). RNA was reversely transcribed with the iScript cDNA synthesis kit (Bio-Rad). Real-time PCR analysis was performed with the KAPA SYBR FAST qPCR MasterMix (Peqlab) on an iCycler IQ system (Bio-Rad). Relative expression levels were normalized to HPRT. For primers, see Supplementary data.

### Agilent microarray data and overlaps
Gene expression profiling was performed with Agilent Whole Genome Microarrays (Agilent's SurePrint G3 Mouse GE 8 × 60K Microarray, Agilent Microarray Design ID 028005). The data analysis was performed using GeneSpring software 11.5 (Agilent Technologies). The cutoffs for differential expression were set as absolute fold-change >2 and a corrected *P*-value of <0.05. GO analysis was performed with the DAVID (http://david.abcc.ncifcrf.gov/). Overlaps were identified with R and compared to the mean of 1000 random overlaps. The significance (*P*-value) from these results was calculated using Fisher's exact test, a hypergeometric probability test to determine the significance between the overlaps (R version 2.15.0 (2012-03-30); function fisher.test).

### Protein isolation, immunoblot analysis and HDAC activity assays
For protein isolation from cultured keratinocytes, cells were scraped and isolated in Frackelton buffer (10 mM Tris–HCl pH7.05, 50 mM NaCl, 30 mM sodium pyrophosphate, 50 mM NaF, 1 % Triton X-100). Protein extracts were vortexed and incubated on ice for 10 min. For protein isolation from tissue, the epidermis was isolated as described above. The tissues were homogenized in Frackelton buffer with ceramic beads (Precellys) for 30 s at 6 m/s with a FastPrep-24 homogenizer (MP Biomedicals). Equal amounts of protein extracts were separated by SDS–PAGE and transferred onto nitrocellulose membranes (Protran, Whatman) according to the standard protocols. For detection, the enhanced chemiluminescence kit (Perkin-Elmer) was used. HDAC activity assays were performed with epidermal protein extracts as previously described (Lagger *et al*, 2002). Frackelton buffer was supplemented with protease inhibitor cocktail (cOmplete, Roche) and aprotinin (Sigma). For antibodies, see Supplementary data.

### ChIP and PCR analysis
Epidermis was chopped, washed with PBS and crosslinked with disuccinimidyl glutarate (2 mM, AppliChem) for 25 min at room temperature (RT). After a PBS washing step, the tissue was crosslinked with formaldehyde (final conc. of 1%) at RT for 10 min. Crosslinking was stopped by addition of 125 mM glycine. Chromatin isolation was done as previously described (Hauser *et al*, 2002). For ChIP, equal amounts of sonicated chromatin were diluted 10-fold and precipitated overnight with antibodies listed in Supplementary data. Chromatin antibody complexes were isolated using protein A-beads or G-beads (Dynabeads©, Invitrogen) and the extracted DNA was used for qRT–PCR. In parallel, PCRs with 1:20 dilutions of genomic DNA (input) were carried out. For primers, see Supplementary data.

### Co-immunoprecipitation assay
Total protein extracts from keratinocytes or epidermal extracts were harvested in Frackelton lysis buffer. Equal amounts of protein were incubated for 2 h at 4°C with 4 µg antibody. Immunoprecipitation was carried out using protein A-beads or protein G-beads O/N at 4°C. Immune complexes were washed with Frackelton lysis buffer. Samples were used for HDAC activity assays or immunoblotting.

### Nuclear extraction

Isolated epidermis was chopped and filtered through a 70 µm cell strainer. After washing, the cell pellet was resuspended in sucrose buffer (0.32 M sucrose, 10 mM Tris–HCl pH 8.0 3 mM $CaCl_2$, 2 mM MgOAc, 0.1 mM EDTA, 0.5% NP-40). Nuclei were isolated by centrifugation and the nuclear pellet was washed twice with sucrose buffer (without NP-40). Then, the pellet was resuspended in low salt buffer (20 mM HEPES pH 7.9, 1.5 mM $MgCl_2$, 20 mM KCl, 0.2 mM EDTA, 25% glycerol) and mixed with an equal amount of high salt buffer (1.5 mM $MgCl_2$, 800 mM KCl, 0.2 mM EDTA, 25% glycerol, 1% NP-40). After shaking for 1.5 h at 4°C the nuclear fraction was obtained as a supernatant after centrifugation.

### Histological and IHC analyses

Tissue samples were fixed overnight in 4% PFA and further embedded in paraffin. Stainings were performed on 4 µm sections. H&E stainings were carried out according to the standard procedure with an ASS1 staining unit (Pathisto). IF stainings were carried out with the Tyramide Signal Amplification Kit (PerkinElmer) according to the manufacturer's instruction. The slides were mounted with DAPI in Vectashield (Vector Laboratories). For IHC stainings with mouse primary antibodies the M.O.M Immunodetection Kit (Vector Laboratories) and for rabbit primary antibodies the VECTASTAIN ABC Kit (Vector Laboratories) were used according to the manufacturer's protocol. Antibody detection was carried out with the DAB Peroxidase Substrate Kit (Vector Laboratories) and sections were counterstained with haematoxylin reagent. For Oil O red staining of SG and for quantification of histology stainings, see Supplementary data.

### Microscopy

IHC stainings, tissue whole mounts and hair samples were captured on a Zeiss stereomicroscope with camera. IHC fluorescence stainings were imaged on a Zeiss LSM Meta 510 confocal microscope.

### Statistical analysis

qRT-PCR and ChIP experiments were evaluated with Microsoft Excel. Immunoblot signal intensities were quantified using the ImageQuant® software and relative protein levels were normalized to β-actin or Lamin B. The significance between groups was determined by the unpaired Student's *t*-test. *P*-values were calculated with the Graph-Pad Prism software and standard deviation (s.d.) is shown. $*P < 0.05$; $**P < 0.01$; $***P < 0.001$; ns = not significant. In Figures 7B and 8B, the data were analysed by a log-rank (Mantel-cox) test. If data were obtained from independent experiments, then controls were set to 1 or 100% and the s.d. of the ratio of the mutant mice in relation to the controls is shown.

For further information of materials (primers, antibodies) and methods, see Supplementary data.

### Supplementary data

Supplementary data are available at *The EMBO Journal* Online (http://www.embojournal.org).

## Acknowledgements

This work was supported by the GEN-AU project 'Epigenetic Regulation of Cell Fate Decisions' (Federal Ministry for Education, Science, and Culture), the Austrian Science Fund (FWF 25807), the University of Vienna Initiativkolleg I031-B, the Herzfelder Family Foundation and the WWTF (LS09-031) to CS. MS acknowledges funding by the EC programs QLG1-CT-2001-00869 and LSHC-CT-2006-037731 (Growthstop), the Austrian Federal Government's GEN-AU program ''Austromouse'' (GZ 200.147/1–VI/1a/2006 and 820966), and Austrian Science Fund grants DK W1212, P18421, P18782, and SFB-23-B1P18421, P18782, and SFB-23-B13. MAM is a fellow of the International PhD program 'Molecular Mechanisms of Cell Signaling' supported by the Austrian Science Fund (W1220). RB was a fellow of the Ernst Schering Foundation. BML was recipient of a Boehringer Ingelheim Fonds fellowship. We are grateful to M Frye, SB McMahon, B Lüscher and R Herbst for plasmids and cell lines. We would like to thank S Lagger, A Sawicka, G Zupkovitz, B Zaussinger, E Pineda, A Hagelkruys, M Rennmayr, I Fischer, H Fischer, N Amberg, F Kern, G Walko, P Andorfer, J Guinea-Viniegra, and T Dechat for technical support, cell lines and antibodies and W Glaser and M Goiser for bioinformatic support. We thank M Baccarini for many fruitful discussions.

*Author contributions*: MW and MAM designed and performed the experiments and co-wrote the manuscript; DM, CF, GM, KM, BML, RB, SW, GB, CM, CH and TM performed experiments. PM provided mice, MS analysed the data and advised on manuscript, CS performed experiments, designed the research, supervised the project, and co-wrote the paper.

## Conflict of interest

The authors declare that they have no conflict of interest.

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
