## [Review Process File · The EMBO Journal]

Manuscript EMBO-2013-85600

Divergent roles of HDAC1 and HDAC2 in the regulation of epidermal development and tumorigenesis

Mircea Winter, Mirjam A. Moser, Dominique Meunier, Carina Fischer, Georg Machat, Katharina Mattes, Beate M. Lichtenberger, Reinhard Brunmeir, Simon Weissmann, Christina Murko, Christina Humer, Tina Meischel, Gerald Brosch, Patrick Matthias, Maria Sibilica and Christian Seiser

Corresponding author: Christian Seiser, Max. F. Perutz Laboratories, University of Vienna

Review timeline:

Submission date:	08 May 2013
Editorial Decision:	03 June 2013
Revision received:	23 September 2013
Additional Editorial Correspondence:	15 October 2013
Additional Author Correspondence:	16 October 2013
Accepted:	21 October 2013

Transaction Report:

Editor: Thomas Schwarz-Romond

1st Editorial Decision

03 June 2013

Thank you very much for submitting your paper on specific and redundant functions of HDAC1/2 in postnatal skin homeostasis for consideration to The EMBO Journal editorial office.

The attached comments outline significant previous work on the subject, while appreciating potential novel insights provided by the allelic series of HDAC1/2-ablation. I do notice that ref#2 appears rather supportive of the study. In contrast, both refs#1 and #3 demand significant further experimentation before being able to support publication here. Specifically:

- much better introduction into affected skin structures and the presumably age-dependent phenotypic changes;
- main focus should be the molecular characterization HDAC1-deficient/HDAC2 monoallelic hair follicles as these appears the most novel observation;
- with myc as the most-likely executing target, the functional link between Sin3A and myc should thoroughly be established, respective direct versus indirect HDAC1/2 targets differentiated;
- with initial implications for a tumor suppressor function already published, again stronger molecular insights, as for instance reduced co-repressor complex formation at target genes in indicated additional tumor backgrounds would have to be provided if you were to decide to maintain this as part of the study.

Though demanding significant further, obviously time-consuming experimentation, these are valuable and constructive critiques to certainly improve general insight(s) and thus relevance of your study.

Assuming that you are in a strong position to address these in a relatively timely manner, we are prepared to offer you the chance for these crucial amendments. Please do not hesitate to contact me to outline feasibility, anticipated timeline or indeed to discuss necessary extensions for our usually limited revision time. (Given our time demands, preferably via E-mail).

I do hope that summarizing/emphasizing crucial requests from our valued referees provides some guidance how to proceed with this project, I am very much looking forward to your response and attention in this matter and remain with best regards.

REFEREE REPORTS:

Referee #1:

In their manuscript Winter and colleagues analyze the functional consequences of dose-dependent deletion of HDAC1 and 2 in the mouse epidermis. In line with previously published data, single deletion of HDAC 1 or 2 does not cause any aberrant phenotype, whereas mice with double deletion are not viable. The authors find that deletion of a single allele of HDAC2 in the absence of HDAC1 causes hair follicle degeneration but increases proliferation and differentiation in the interfollicular epidermis (IFE) and sebaceous glands (SG). As a mechanism the authors suggest that the formation of co-repressor complexes at genes involved in driving IFE proliferation and differentiation is reduced. Finally, the authors provide evidence that HDAC1 may act as a tumor suppressor in mammalian epidermis.

General and specific HDAC-inhibitors are widely investigated as possible treatment for various human diseases including cancer. Therefore, studies investigating the precise role of specific HDACs in developing and adult tissues are highly important to determine potential applications and side effects of HDAC-inhibitors. However, the roles of HDAC1 and 2 have been widely studied in various tissues including the skin (LeBoeuf et al., 2010). General inhibition of class I and II mammalian histone deacetylases by trichostatin A has been shown to increase proliferation and differentiation in the interfollicular epidermis and sebaceous glands (Frye et al., 2007). Finally, the specific roles for HDAC 1 and 2 in tumor development and its relation to Myc and p53 have been recently shown in thymic lymphomas (Heideman et al., 2013). In light of the published data, the manuscript falls short in providing a mechanism why hair follicle and IFE / SG are differently affected by deletion of a single allele of HDAC2 in the absence of HDAC1.

The authors might want to consider restructuring their manuscript. The description of the skin phenotype (figures 2-5) is confusing because it lacks structure and focus. A suggestion may be:

1. Description of the phenotype structured by age from P5 to P150
2. Molecular characterization of Hdac1^{-/-}Hdac2^{+/-} hair follicles since this is (i) the most obvious phenotype, (ii) can explain the proliferation effects on both interfollicular epidermis and sebaceous glands, and (iii) is a novel observation.
3. Depletion / degeneration of hair follicles and analyses of bulge cells due to enhanced contribution of hair follicle cells to IFE and SG.
4. Molecular characterization of Hdac1^{-/-} Hdac2^{+/-} IFE and SG.

Specific Comments:

1. The results shown in figure 1 can be summarized as supplementary information because the relevance for the overall manuscript is minor and the redundant function of HDAC1 and HDAC2 in the developing epidermis is not novel (LeBoeuf et al., 2010). The authors should also refer to the published data in this section.
2. Figure 2: It is unclear why the authors did not include data shown in figure S3 into figure 2. The

phenotypic description, shown in figures S3C-H are novel findings since the double knockout is not viable after birth.

3. Figure 2B can be considered as supplementary information.
4. In line with my comments above, referring to hair follicles and IFE in figure 3 is confusing because the authors then focus again on the hair follicle in figure 4.
5. Figure 3A: The authors conclude that "the back skin was thinner,". This conclusion cannot be drawn from mice at P35. Since the first hair cycle is synchronized, the majority of mice should display hair follicles at the end of anagen beginning of catagen. The thickness of the epidermis depends on the hair cycle stage (always thicker in anagen) and the skin in Hdac1^{-/-} Hdac2^{-/+} mice might simply be thinner because it is not in anagen but already in catagen. Thus, this statement needs some quantification and a better determination of the hair cycle state in Hdac1^{d/d} Hdac2^{d/+} epidermis is at P35. In other words, do Hdac1^{-/-} Hdac2^{-/+} mice enter catagen prematurely?
6. Figure 4B: The authors should consider alternative explanations for the observed phenotypes. The skin at P5 might be thinner because the mice show delayed entry into anagen, which is in line with the observation the mice are much smaller at birth and in general underdeveloped. It is interesting though that Hdac1^{-/-} Hdac2^{-/+} epidermis is able to enter the first anagen and may not be able to establish a telogen hair follicle (P20). Due to the degeneration of the hair follicle, Hdac1^{-/-} Hdac2^{-/+} epidermis may never be able to enter the second anagen.
7. An alternative explanation for the phenotype at P20 is a prolonged anagen phase due to the "wounding and scaling" effect cause by Hdac1^{-/-} Hdac2^{-/+}. Hair follicles stay or enter anagen in response to wounding (Ansell et al., 2011; Ito et al., 2005).
8. A mis-regulation in catagen would explain the phenotype at P35 and the loss of Gata3 (Fig. 4E). Catagen is indicated by a high rate of apoptosis and low rate of proliferation. An induction of p53 followed by apoptosis would fit in well into published observations in the IFE by (LeBoeuf et al., 2010).
9. The failure to differentiate into specific hair lineages should be confirmed by staining for markers such as K31, K72 etc.
10. The increase in proliferation and thickness in the IFE and SG at expense of the hair follicle is interesting and indicates that bulges stem cells do not contribute to hair follicle regeneration but may be still able to contribute to IFE regeneration. This would also explain the reduction in bulge stem cell markers but increase in Lgr6-positive progenitors. A simple explanation may be that the hair follicle responds to p53 with apoptosis while the IFE is highly resistant to p53-induced apoptosis in the absence of UV radiation for instance.
11. In line with point 10 and with the observation of scaling and spontaneous wounds, it would be interesting to know whether Hdac1^{-/-} Hdac2^{-/+} mice display wound healing defects.
12. Figure 5D: The quantification of the label-retaining cells does not allow any conclusion with regard to slow cycling stem cell populations in its presented form. The data indicate that proliferation in the hair follicle is reduced and increased in IFE at the time of labeling. This means that the up-take of BrdU is different in wild-type and transgenic mice at the beginning of the experiment. The authors either need to show that they obtain similar BrdU labeling at the pulse or need to revise the conclusions from this experiments.
13. As potential mechanism, the authors suggest a reduced formation of co-repressor complexes in Hdac1^{-/-} Hdac2^{-/+} epidermis on genes involved in proliferation and differentiation. It is unclear, how this can explain the specificity observed in skin (interfollicular epidermis versus hair follicle).
14. If co-repressor complexes form to a lesser extent, any repressed gene should be re-activated to a certain extent. How does the occurrence of HDAC1 and 2 changes at genes that are not differentially expressed in Hdac1^{-/-} Hdac2^{-/+} versus Hdac1^{-/-} epidermis?
15. If the hypothesis is correct, the authors should also find reduced occurrence of Sin3A and MTA2 but increased presence of transcriptional co-activators at these genes.
16. The reasoning for performing the tumor experiments and the contribution of the data to the overall hypothesis is unclear. Since only data from the single knockout mice are shown, at the very minimum the authors need to confirm that co-repressor complex formation at specific genes (see figure 6D-E and 7) is reduced in in a RAS background when only HDAC1 is deleted.

Ansell, D.M., Klopper, J.E., Thomason, H.A., Paus, R., and Hardman, M.J. (2011). Exploring the "hair growth-wound healing connection": anagen phase promotes wound re-epithelialization. *The Journal of investigative dermatology* 131, 518-528.

Frye, M., Fisher, A.G., and Watt, F.M. (2007). Epidermal stem cells are defined by global histone modifications that are altered by Myc-induced differentiation. *PLoS ONE* 2, e763.

Heideman, M.R., Wilting, R.H., Yanover, E., Velds, A., de Jong, J., Kerkhoven, R.M., Jacobs, H., Wessels, L.F., and Dannenberg, J.H. (2013). Dosage-dependent tumor suppression by histone deacetylases 1 and 2 through regulation of c-Myc collaborating genes and p53 function. *Blood* 121, 2038-2050.

Ito, M., Liu, Y., Yang, Z., Nguyen, J., Liang, F., Morris, R.J., and Cotsarelis, G. (2005). Stem cells in the hair follicle bulge contribute to wound repair but not to homeostasis of the epidermis. *Nat Med* 11, 1351-1354.

LeBoeuf, M., Terrell, A., Trivedi, S., Sinha, S., Epstein, J.A., Olson, E.N., Morrisey, E.E., and Millar, S.E. (2010). Hdac1 and Hdac2 act redundantly to control p63 and p53 functions in epidermal progenitor cells. *Developmental cell* 19, 807-818.

Referee #2:

This manuscript addresses the role of HDAC1/2 in postnatal skin homeostasis. While previous work has illustrated a role for HDAC1/2 in the development of the epidermis prior to birth, the role of HDAC1/2 in postnatal skin homeostasis has not been reported. The authors generate mice lacking HDAC1 and 1 allele of HDAC2 and find a dramatic alopecia phenotype in the skin of adult mice. Further analysis of hair shafts of HDAC1null; HDAC2 het mice reveals a defect in hair shaft structure. They find alterations in the interfollicular epidermis, hyperproliferation and decreased differentiation and sebaceous gland hyperplasia.

The data within this manuscript make several interesting and novel points. First, they show that while some functions of HDAC1 and HDAC2 are redundant in the skin, HDAC1 and HDAC2 do have some independent functions in postnatal skin homeostasis. The data also indicate a function for HDAC1/2 in the regulation of epidermal proliferation/differentiation in the epidermis, hair follicle and sebaceous gland. Finally, the authors use several assays to link HDAC function with c-myc expression. These convincing and intriguing data make a significant advance in the field of epidermal and chromatin biology and thus, are worthy of publication in EMBO J. I have a few concerns discussed below but overall this is an excellent manuscript.

1. The data thoroughly and convincingly describe the phenotypic characteristics of HDAC1null; HDAC2 het mice in all three epithelial lineages, the epidermis, hair follicle and sebaceous gland. Mechanistically, the authors provide evidence that these phenotypes may arise from the direct interaction of HDAC1/2 with the promoters of several epidermal differentiation genes and its interaction with Sin3a. Since c-myc can alter epidermal and sebaceous gland homeostasis, these mechanistic data address 2 of the 3 epidermal lineages. However, the authors do not analyze whether HDAC1/2 occupies genes that are altered in the hair follicles of HDAC1null; HDAC2 het mice, such as Hoxc13. These data would link the hair phenotypes to HDAC function

2. The tumor data in the final figure seem disjointed from the rest of the manuscript since the enhanced tumor phenotype exists in HDAC1 null mice on their own, suggesting that the homeostasis phenotype is distinct from HDAC1's role as a tumor suppressor. I recommend removing this figure and exploring this portion of the data as a separate manuscript.

3. In Figure 6, the authors compare mRNA changes in the skin of HDAC1null; HDAC2 het mice compared to changes in the skin of Sin3a null mice and the skin of mice overexpressing c-Myc. To determine if the shared overlap is significantly significant, the authors should perform a hypergeometric probability test.

4. In Fig. 3. The age of mice should be indicated on figure or in legend for the histology and immunofluorescence images.

5. The authors describe the experiments resulting from HDAC1null; HDAC2 het mice as illustrating "uncompensated HDAC1 function". While the authors do show that HDAC2 is elevated upon HDAC1 deletion, it is unclear whether uncompensated function is actually the cause of the particular phenotype. Describing the function as HDAC function more generally is more accurate.

6. In the discussion, the authors should link the sebaceous gland phenotype to c-myc function as described by Arnold 2001, Lo Celso, 2008.

Referee #3:

In this paper, Winter et al. describe the affect of loss of Hdac1 and Hdac2 from the mouse epidermis. Hdac1/2 2KO has already been described as having a strong phenotype during skin development, with impaired stem cell proliferation and differentiation and loss of barrier function of the epidermis (LeBoeuf et al. 2011). It had also been described that the single Hdac1 cKO and Hdac2 cKO do not have a phenotype (LeBoeuf et al. 2011). In this paper the authors describe that the specific combination of Hdac1 cKO and Hdac2 heterozygous (het) affects skin morphogenesis in adult mice, with increased proliferation and differentiation in the interfollicular epidermis and degeneration of the hair follicle. This phenotype is somewhat similar to previously described phenotype for the overexpression of cMyc in the epidermis (Frye et al 2003). While the phenotype is very interesting, it could be better described in terms of its progression as it appears that it becomes more sever with time, as the mice age and there are also indications of increased stress and injury in the skin an potential inflammation. The authors found that the phenotype is due, at least in part to the increase levels of cMyc protein as well as increase expression of other genes involved in skin morphogenesis and proliferation. Thus, Hdac1 seems to have a role in preventing excessive proliferation in the adult skin, which can only be compensated in part by Hdac2. This role of Hdac1 as a tumor suppressor gene is further substantiated by genetic interaction in a mouse model for tumorigenesis. However, Hdac1 also seems to have a role in regulation inflammation, which needs to be better characterized. The results are interesting and novel. However, the authors should better discuss a model for the tissue specific role of Hdac1 as a tumor suppressor gene and which genes repressed directly regulated by Hdac1 might be participating directly or indirectly in this role.

Major comments:

- In the introduction, the interfollicular epidermis and the hair follicle should be introduced in more detail, specifically mentioning the structures that are relevant for the understanding of the paper. Also, the known role of Hdac1/2 in epidermal should be stated more clearly.

- Figure1:

- The IF stainings for Hdac1 and Hdac2 in Fig1 and Fig S1 should be co-stained with a basal marker such as K14 or 4 and a suprabasal maker to allow for a better interpretation of the expression pattern of Hdac1/2.

- The age of all mice shown (Fig1A, SFig2G) as well as the age of the mice for the skin samples (Fig1B, FigS2A,B,H) should be indicated in all panels. This should be done for all mice shown and all skin histology panels in all subsequent figures.

- The H&E staining or the IF presented in Figs2A and B are more informative of the correct development of the epidermis in Hdac1cKO and Hdac2cKO and should be included in Fig1 or replace Fig1A.

- The methodology to quantify Ki67 positive cells (FigS2C) and epidermal thickness (FigS2D) should be clearly described in the materials and methods and statistical analysis should be preformed for all quantifications and the P values should be indicated in the figure legends. This should be done for all quantifications in all subsequent figures.

- Statistical analysis should be preformed for the quantification of Fig1C and D and the P values should be indicated in the figure legends, including for FigS2E,D.

- Figure2:

- Again, the IF stainings for Hdac1 and Hdac2 in Fig2B should be co-stained with basal and suprabasal makers to allow for a better interpretation of the expression pattern of Hdac1/2.

- Statistical analysis should be preformed for the quantification of Fig2C and D and Fig24A,B and the P values should be indicated in the figure legends.

- To fundament their claim that certain histone acetylation marks are increased in the epidermis of Hdac1cKO Hdac2het, the authors should quantify the immunoblots of FigS4C and perform statistical analysis.

- Figure3:

- The age of the mice analyzed in Fig3A, B and D should be indicated in the panels.
- The claim that there is an enlargement of the basal and spinous layer in the Hdac1cKO Hdac2het is not very clearly illustrated in Fig3B and the authors should present a more convincing example. Again, the description of how the quantification on the thickness of the epidermis was performed (Fig3C) should be clearly described.
- The graphs presented, such as in Fig3E should clearly indicated whether it is protein levels of mRNA that is being shown, making the interpretation of the figures more immediate. This is should be indicated in all graphs showing either protein levels or mRNA expression levels in all figures

- Figure4:

- Again, the authors should include the age of all samples shown in Fig4 as well as clear description of how the SG gland sizes was quantified and the P values for the statistical tests.

-Figure5:

- The age of the skin analyzed for proliferation in Fig5A and S4E is not indicated. Also, the phenotype of the Hdac1cKO Hdac2het mice appears to become more severe with time (Fig S3D-H). In Hdac1/2 2KO there is impairment of barrier function of the skin. The data presented in this paper does not indicate whether there is also impairment of barrier function in the Hdac1cKO Hdac2het - this must be analyzed in the postnatal mice. Impairment of the barrier function of the skin has been shown to induce inflammation in the skin, which has been associated with increased proliferation and expression of differentiation markers. Since there is both an increase in proliferation but also an increase in expression of inflammation related makers such as K6, K16, S100a8 and S100a9A in the Hdac1cKO Hdac2het mice, the authors must address whether there is inflammation in these mutants. . A time course analysis should be preformed in neonatal mice to determine which defects can be detected earlier and whether there is an increase in hyperplroliferation in the epidermis as the tissue begins to show more signs stress and damage. Also, the authors should investigate whether there is infiltration of immune cells in the epidermis with markers such as CD45.
- The authors should inhibit inflammation with the anti-inflammatory drug dexamethasone (Perez-Moreno et al, 2006) to address which aspects of the Hdac1cKO Hdac2het phenotype is due to inflammation and which might be due to increased cMyc protein.
- The claim that Hdac1cKO Hdac2het mice have defects in lineage determination and that there is a mobilization of epidermal stem cells should be carefully discussed. The data presented does not exclude that there is mobilization of the HF stem cell to the interfollicular epidermis as a result of wounding of the epidermis.
- The methodology for quantifying the signal from in situ hybridization and the BrdU+ cells in the epidermis should be described clearly in the materials and methods.
- The expression level of cMyc immunoblot (Fig5G) should be quantified.

- Figure6:

- What are the 79 genes that are affected in the Hdac1cKO and that cannot be compensated by Hdac2? Are these relevant for the interpretation of the phenotype of Hdac1cKO Hdac2het versus the Hdac1/2 2cKO? Are these genes related to inflammation? Also, does Hdac1 bind to the promoter of these genes directly? This could be tested with the CHIP already performed and could be very informative on why the Hdac1cKO Hdac2het and the Hdac1/2 2cKO mice have different phenotypes.
- The effect of loss of deactylase activity in Hdac1cKO Hdac2het mice appears to only have an indirect effect on cMyc or at least only affect its post-translational stability. What is the nature of the functional link between Sin3A and cMyc? Could the instability of the Hdac1 repressor complex with Sin3A and MTA2 explain the persistence of cMyc protein? Does cMyc interact directly with Hdac1 repressive complexes? The authors should better discuss their hypothesis.
- The immunoblots in Fig6 E and G should be quantified to support the claims made by the authors on the change in the levels of Sin3A and MTA2.

- Figure7:

- How generalizable is the absence of complete compensation by Hdac2 in the Hdac1cKO Hdac2het mice? The authors only show two genes (Sprr2h and Epgn) in Fig7B. It will be important to understand whether other genes behave like this. The authors should present other examples of genes that have Hdac1 directly bound to their promoter. Furthermore, might any of the 79 genes that

are not at all compensated by Hdac2 in the Hdac1cKO play a role in the phenotype observed? While the deregulation alone of these genes might not have a phenotype, they could interact with other genes that are deregulated in the Hdac1cKO Hdac2het mice. This might be an interesting point for discussion. Finally, can the authors comment on how many of the deregulated genes have Hdac1 directly bound at their promoter? This could help explain direct and indirect effects of loss of Hdacs and therefore their role in promoting hyperproliferation or suppressing it, depending on the context.

-Figure8

- In continuation of the comment above, the authors should discuss why Hdac1 might have a tumor suppressor role and why Hdac2 does not. What are the relevant targets that might justify this difference?

- Inflammation has been shown to have a link with tumor development. The authors must inhibit inflammation in their K5-SOS Hdac1cKO mice to address whether Hdac1 has a tumor suppressor independent of inflammation.

- Again, information on age of the mice is missing from the figure, as well as quantification for some the immunoblots and statistics.

Minor comments:

- Fig3A has a white arrow that is not described in the figure legend neither in the text.

1st Revision - authors' response

23 September 2013

Referee #1:

In their manuscript Winter and colleagues analyze the functional consequences of dose-dependent deletion of HDAC1 and 2 in the mouse epidermis. In line with previously published data, single deletion of HDAC 1 or 2 does not cause any aberrant phenotype, whereas mice with double deletion are not viable. The authors find that deletion of a single allele of HDAC2 in the absence of HDAC1 causes hair follicle degeneration but increases proliferation and differentiation in the interfollicular epidermis (IFE) and sebaceous glands (SG). As a mechanism the authors suggest that the formation of co-repressor complexes at genes involved in driving IFE proliferation and differentiation is reduced. Finally, the authors provide evidence that HDAC1 may act as a tumor suppressor in mammalian epidermis.

General and specific HDAC-inhibitors are widely investigated as possible treatment for various human diseases including cancer. Therefore, studies investigating the precise role of specific HDACs in developing and adult tissues are highly important to determine potential applications and side effects of HDAC-inhibitors. However, the roles of HDAC1 and 2 have been widely studied in various tissues including the skin (LeBoeuf et al., 2010). General inhibition of class I and II mammalian histone deacetylases by trichostatin A has been shown to increase proliferation and differentiation in the interfollicular epidermis and sebaceous glands (Frye et al., 2007). Finally, the specific roles for HDAC 1 and 2 in tumor development and its relation to Myc and p53 have been recently shown in thymic lymphomas (Heideman et al., 2013). In light of the published data, the manuscript falls short in providing a mechanism why hair follicle and IFE / SG are differently affected by deletion of a single allele of HDAC2 in the absence of HDAC1.

Reorganization of the manuscript:

The authors might want to consider restructuring their manuscript. The description of the skin phenotype (figures 2-5) is confusing because it lacks structure and focus. A suggestion may be:

1. *Description of the phenotype structured by age from P5 to P150*
2. *Molecular characterization of Hdac1^{-/-}Hdac2^{-/+} hair follicles since this is (i) the most obvious phenotype, (ii) can explain the proliferation effects on both interfollicular epidermis and*

sebaceous glands, and (iii) is a novel observation.

3. *Depletion / degeneration of hair follicles and analyses of bulge cells due to enhanced contribution of hair follicle cells to IFE and SG.*

4. *Molecular characterization of Hdac1^{-/-} Hdac2^{+/-} IFE and SG.*

We thank the reviewer for the constructive criticism and have reorganized the results accordingly. In brief, we have described the phenotypes of *Hdac1^{Δ/Δep} Hdac2^{Δ/+ep}* mice according to the age in Figure 1. The molecular characterization of the hair follicles and the hair cycle of the mutant mice is shown now in Figure 2 including new data on the degeneration and apoptosis. The description of hyperkeratosis and hypertrophy of sebaceous glands follows in Figure 3 and the molecular characterization of bulge cells, hair follicles and IFE is shown in Figure 4. We are convinced that the reorganization makes the description of lineage specific effects of HDAC1/HDAC2 depletion in the epidermis more clear.

Specific Comments:

1. *The results shown in figure 1 can be summarized as supplementary information because the relevance for the overall manuscript is minor and the redundant function of HDAC1 and HDAC2 in the developing epidermis is not novel (LeBoeuf et al., 2010). The authors should also refer to the published data in this section.*

We now show the data with individual knockouts of HDAC1 and HDAC2 in the epidermis in the Supplementary Figure 1 and cite the paper by LeBoeuf et al. in the corresponding section of the results.

2. *Figure 2: It is unclear why the authors did not include data shown in figure S3 into figure 2. The phenotypic description, shown in figures S3C-H are novel findings since the double knockout is not viable after birth.*

As suggested we show now the data describing age-dependent phenotypes of *Hdac1^{Δ/Δep} Hdac2^{Δ/+ep}* mice including new data showing spontaneous tumor development in older mice in Figure 1.

3. *Figure 2B can be considered as supplementary information.*

The Figure is shown now as part of Supplementary Figure 1.

4. *In line with my comments above, referring to hair follicles and IFE in figure 3 is confusing because the authors then focus again on the hair follicle in figure 4.*

This has been changed accordingly; the hair follicle data are shown now in Figure 2 (see above).

5. *Figure 3A: The authors conclude that "the back skin was thinner,". This conclusion cannot be drawn from mice at P35. Since the first hair cycle is synchronized, the majority of mice should display hair follicles at the end of anagen beginning of catagen. The thickness of the epidermis depends on the hair cycle stage (always thicker in anagen) and the skin in Hdac1^{-/-} Hdac2^{+/-} mice might simply be thinner because it is not in anagen but already in catagen. Thus, this statement needs some quantification and a better determination of the hair cycle state in Hdac1^{d/d} Hdac2^{d/+} epidermis is at P35. In other words, do Hdac1^{-/-} Hdac2^{+/-} mice enter catagen prematurely?*

6. *Figure 4B: The authors should consider alternative explanations for the observed phenotypes. The skin at P5 might be thinner because the mice show delayed entry into anagen, which is in line with the observation the mice are much smaller at birth and in general underdeveloped. It is interesting though that Hdac1^{-/-} Hdac2^{+/-} epidermis is able to enter the first anagen and may not be able to establish a telogen hair follicle (P20). Due to the degeneration of the hair follicle, Hdac1^{-/-} Hdac2^{+/-} epidermis may never be able to enter the second anagen.*

7. *An alternative explanation for the phenotype at P20 is a prolonged anagen phase due to the "wounding and scaling" effect cause by Hdac1^{-/-} Hdac2^{+/-}. Hair follicles stay or enter anagen in response to wounding (Ansell et al., 2011; Ito et al., 2005).*

8. *A mis-regulation in catagen would explain the phenotype at P35 and the loss of Gata3 (Fig. 4E). Catagen is indicated by a high rate of apoptosis and low rate of proliferation. An induction of p53 followed by apoptosis would fit in well into published observations in the IFE by (LeBoeuf et al., 2010).*

We fully agree with the reviewer. As suggested we have quantified hair follicle length at different hair cycle stages. Interestingly, hair follicle development in *Hdac1^{Δ/Δep} Hdac2^{Δ/+ep}* mice is impaired already during morphogenesis. Mutant HF are able to enter morphogenesis and as suggested by the reviewer *Hdac1^{Δ/Δep} Hdac2^{Δ/+ep}* HF seem to be unable to establish a telogen HF, since at P18 HF are longer. These data are shown in Figs. 2A-C. We cannot completely exclude that that a “wounding and scaling” effect negatively affects hair cycle progression. However it is unlikely since wounding is restricted to the tail and not detectable on all areas of the mutant back skin. Furthermore, the mutant HF are not able to enter the next round of anagen (P35), since the HF length remains unchanged after telogen.

Importantly, we detect p53 expression already during morphogenesis of mutant HF resulting in apoptosis and HF atrophy, shown in Figure 2. As highlighted by the reviewer this is in line with the finding that *HDAC1/2* are necessary to suppress active p53 in the epidermis to prevent proliferative defects and apoptosis (LeBoeuf et al., 2010). This is a very important finding that helps to explain the lineage-specific effects in *Hdac1^{Δ/Δep} Hdac2^{Δ/+ep}* mice (see also below under Point 10).

9. *The failure to differentiate into specific hair lineages should be confirmed by staining for markers such as K31, K72 etc.*

We have analyzed the following markers GATA3, Lhx2, S100A3, Hoxc13, Msx2, keratin 31 and keratin 71. In agreement with the hair/HF phenotype and the absence of CD34 in HF we find deregulation of GATA3, Lhx2, S100A3, Hoxc13, Msx2 as shown in Figure 2. In addition, we performed IF stainings of specific hair keratins. Staining for K71 (Henle, Huxley and cuticle layer of the hair follicle inner root sheath) and K31 (lower hair cortex) did not reveal a change in the differentiation state of the respective layers, indicating that the phenotype is not due to a differentiation defect of the progenitor cells (see attached Figure).

10. *The increase in proliferation and thickness in the IFE and SG at expense of the hair follicle is interesting and indicates that bulges stem cells do not contribute to hair follicle regeneration but may be still able to contribute to IFE regeneration. This would also explain the reduction in bulge stem cell markers but increase in Lgr6-positive progenitors. A simple explanation may be that the hair follicle responds to p53 with apoptosis while the IFE is highly resistant to p53-induced apoptosis in the absence of UV radiation for instance.*

The reviewer raises a crucial point. As discussed under Point 8 we found indeed p53 expression both in the HF and the IFE of *Hdac1^{Δ/Δep} Hdac2^{Δ/+ep}* mice. The analysis of p53 expression and apoptosis at P5 (Fig. 2D, Suppl. Figs S3B and C) indicates that the HF of mutant mice is sensitive to p53-induced apoptosis while the IFE is more resistant. This is a very important point that helps to explain the lineage-specific effects in *Hdac1^{Δ/Δep} Hdac2^{Δ/+ep}* mice (see also Point 8).

11. *In line with point 10 and with the observation of scaling and spontaneous wounds, it would*

be interesting to know whether Hdac1^{-/-} Hdac2^{-/+} mice display wound healing defects.

As suggested by the reviewer we have performed wound healing assays. Strikingly, these experiments revealed a wound healing defect in mutant mice. The wounds do not close properly, show massive keratin deposition around the wound margins leading to lesions that resemble keratoacanthomas. These new data shown in Suppl. Fig. S4 are in line with the observed hyperproliferation in the *Hdac1^{Δ/Δep} Hdac2^{Δ/+ep}* epidermis and the spontaneous formation of tumors in the mutant mice.

12. *Figure 5D: The quantification of the label-retaining cells does not allow any conclusion with regard to slow cycling stem cell populations in its presented form. The data indicate that proliferation in the hair follicle is reduced and increased in IFE at the time of labeling. This means that the up-take of BrdU is different in wild-type and transgenic mice at the beginning of the experiment. The authors either need to show that they obtain similar BrdU labeling at the pulse or need to revise the conclusions from this experiments.*

As suggested by the reviewer we have performed a short pulse labeling experiment with mutant and wildtype mice at P5 (shown in Fig. 3H). As expected based on other proliferation markers we detected increased proliferation in the IFE of *Hdac1^{Δ/Δep} Hdac2^{Δ/+ep}* epidermis and reduced proliferation in the HF of mutant epidermis. The cells that are labeled after a single short BrdU pulse are cycling cells within the epidermis and quickly lose the label due to continuous cycling. Cells with higher proliferation rates lose the label even faster. The number of pulse-labeled cells therefore allows no conclusions with respect to labeling of slow cycling stem cells in wildtype and mutant epidermis.

Slowly cycling stem cells within the hair follicle bulge, which can produce progeny that differentiate into all the hair follicle lineages are labeled with BrdU treatment for several days. During the chase the label of all cycling cells disappears. If there is less labeling of stem cells in the hair follicle bulge of mutant mice, the significant increase in LRCs in the IFE of *Hdac1^{Δ/Δep} Hdac2^{Δ/+ep}* mice is even more striking since there is usually only little presence of LRCs in wildtype epidermis (see for instance Terskikh et al. 2011) after this time period. The shift from the HF to the IFE and SG lineage, observed in *Hdac1^{Δ/Δep} Hdac2^{Δ/+ep}* epidermis was also seen in the c-Myc overexpressing mice as well it is the case in the Sin3A knockout mice, both resembling the phenotype of *Hdac1^{Δ/Δep} Hdac2^{Δ/+ep}* mice. The consequences of the shift becomes also clear in the profile of the gene expression array, which shows a defective HF SC compartment, as seen by the reduction in *Lhx2*, *CD34* and other HF SC genes. However, given the findings of differential sensitivity of the epidermal lineage towards p53-induced apoptosis we have toned down our conclusions.

13. *As potential mechanism, the authors suggest a reduced formation of co-repressor complexes in Hdac1^{-/-} Hdac2^{-/+} epidermis on genes involved in proliferation and differentiation. It is unclear, how this can explain the specificity observed in skin (interfollicular epidermis versus hair follicle).*

14. *If co-repressor complexes form to a lesser extent, any repressed gene should be re-activated to a certain extent. How does the occurrence of HDAC1 and 2 changes at genes that are not differentially expressed in Hdac1^{-/-} Hdac2^{-/+} versus Hdac1^{-/-} epidermis?*

The effect of impaired co-repressor function depends not only on the loss of the repressor but is also strongly affected by the presence of transcriptional activators such as c-Myc and p53 and gain of their (cell type specific) expression activity. Accordingly, the impaired co-repressor function of Sin3A results in de-repression of transcriptional programs regulating for instance EDC genes. On the other hand, the Sin3A complex with its catalytic components HDAC1/HDAC2 controls also the abundance and activity of c-Myc thereby attenuating its (proto-)oncogene function. This is shown by the significant overlap of deregulated genes in *Hdac1^{Δ/Δep} Hdac2^{Δ/+ep}* and c-Myc overexpressing epidermis (shown in Figure 5C). We have explained this model now in more detail in the Discussion (pages 19/20). Finally, given the differential sensitivity of HFs and IFE to p53 activity in HFs, p53 can exert its apoptotic program in a specific epidermal lineage. Unpublished data from ChIP-seq experiments performed in our group show that only a subfraction of genes that are associated with HDAC1/HDAC2 co-repressors respond to loss of HDAC proteins or activity by re-activation. The molecular basis for the differential expression patterns is currently unclear but might include differences in the recruitment of histone acetyltransferases and chromatin remodeling machineries and presence/absence of other histone marks such as histone H3K4me3 or histone H3S10ph.

15. *If the hypothesis is correct, the authors should also find reduced occurrence of Sin3A and MTA2 but increased presence of transcriptional co-activators at these genes.*

Based on the fact that the mutant mice partially phenocopy conditional Sin3A knockout mice we have focused on Sin3A and have performed additional ChIP experiments. As shown in Figure 6 and Supplementary Figure 7 Sin3A is indeed reduced in its presence in *Hdac1 Δ/Δ ep Hdac2 $\Delta/+ep$* epidermis. Together with the loss of HDAC1 and the reduced presence of HDAC2 this results in increased local histone acetylation at target genes. These new data are also shown in Figure 6 and Supplementary Figure 7.

16. *The reasoning for performing the tumor experiments and the contribution of the data to the overall hypothesis is unclear. Since only data from the single knockout mice are shown, at the very minimum the authors need to confirm that co-repressor complex formation at specific genes (see figure 6D-E and 7) is reduced in in a RAS background when only HDAC1 is deleted.*

Based on new data such as the discovery of spontaneous tumors in older mutant mice, the results from wound healing assays suggested by the reviewer we have performed additional experiments on the tumor suppressor function of HDAC1. This was also strongly suggested by the editor. Analysis of HDAC1/HDAC2 co-repressors in HDAC1-deficient SOS tumors revealed a significantly reduced activity for these complexes (Figure 7H). This is in contrast to HDAC1 deficient epidermis, which shows no changes in co-repressor functions (Suppl. Figure S1F) suggesting that under oncogenic stress HDAC2 cannot compensate for the loss of HDAC1. Importantly reduced co-repressor activity is connected with increased c-Myc protein levels, increased proliferation and accelerated development of HDAC1-deficient SOS tumors (Figure 7). In contrast loss of HDAC2 has no effect on co-repressor activity, c-Myc expression, proliferation or tumorigenesis (Figure 8). These data perfectly complement the data showing hyperproliferation in *Hdac1 Δ/Δ ep Hdac2 $\Delta/+ep$* but not *Hdac1 $\Delta/+ep Hdac2 Δ/Δ ep$* mice and provide evidence for a novel non-chromatin associated function of HDAC1 as component of the Sin3A complex in the attenuation of the activity of the proto-oncogene c-Myc in the epidermis.

Referee #2:

1. *The data thoroughly and convincingly describe the phenotypic characteristics of HDAC1null; HDAC2 het mice in all three epithelial lineages, the epidermis, hair follicle and sebaceous gland. Mechanistically, the authors provide evidence that these phenotypes may arise from the direct interaction of HDAC1/2 with the promoters of several epidermal differentiation genes and its interaction with Sin3a. Since c-myc can alter epidermal and sebaceous gland homeostasis, these mechanistic data address 2 of the 3 epidermal lineages. However, the authors do not analyze whether HDAC1/2 occupies genes that are altered in the hair follicles of HDAC1null; HDAC2 het mice, such as Hoxc13. These data would link the hair phenotypes to HDAC function*

As suggested by the reviewer we have included Hoxc13 and Msx2 in addition to marker genes for the epidermis (*Spr2h, S100a8, S100a9, Lce3b, Epgn, Ada*) and the sebaceous gland (*Klk6*) in our ChIP analysis and have analyzed presence of HDAC1, HDAC2 and histone acetylation. Our data (Suppl. Figure S7B) show that both HDAC1 and HDAC2 are present on the promoters of these genes in wildtype epidermis. In *Hdac1 Δ/Δ ep Hdac2 $\Delta/+ep$* epidermis loss of HDAC1 coincides with a reduced histone acetylation at the promoter and lower expression of these hair follicle regulator genes. A similar positive function for HDAC1/HDAC2 in transcriptional regulation was previously documented by Wang et al., 2009, Cell 138:1019-31 in human CD4+ T cells.

2. *The tumor data in the final figure seem disjointed from the rest of the manuscript since the enhanced tumor phenotype exists in HDAC1 null mice on their own, suggesting that the homeostasis phenotype is distinct from HDAC1's role as a tumor suppressor. I recommend removing this figure and exploring this portion of the data as a separate manuscript.*

As discussed above Reviewer 1 Point 16 we were strongly encouraged by the editor to keep and extend the tumor part of the manuscript. We have discovered spontaneous tumor formation in older mutant mice and observed delayed wound healing with formation of lesions in *Hdac1 Δ/Δ ep Hdac2 $\Delta/+ep$* epidermis. During the revision we have produced a set of new data that fits very well to the

development part of the manuscript providing evidence for a novel function for HDAC1 as tumor suppressor in the epidermis.

3. *In Figure 6, the authors compare mRNA changes in the skin of HDAC1null; HDAC2 het mice compared to changes in the skin of Sin3a null mice and the skin of mice overexpressing c-Myc. To determine if the shared overlap is significantly significant, the authors should perform a hypergeometric probability test.*

As suggested we have performed a hypergeometric probability test with random overlaps for the comparison of gene expression profiles shown in Figure 5C.

4. *In Fig. 3. The age of mice should be indicated on figure or in legend for the histology and immunofluorescence images.*

Where applicable the age of the respective mice is now indicated either on the figure (for mice) or in the legend (histology).

5. *The authors describe the experiments resulting from HDAC1null; HDAC2 het mice as illustrating "uncompensated HDAC1 function". While the authors do show that HDAC2 is elevated upon HDAC1 deletion, it is unclear whether uncompensated function is actually the cause of the particular phenotype. Describing the function as HDAC function more generally is more accurate.*

Yes, we agree. We have changed the corresponding sentences.

6 *In the discussion, the authors should link the sebaceous gland phenotype to c-myc function as described by Arnold 2001, Lo Celso, 2008.*

We discussed the link between c-Myc and the SG phenotype in mutant mice in the Discussion section on page 19.

Referee #3:

Major comments:

- *In the introduction, the interfollicular epidermis and the hair follicle should be introduced in more detail, specifically mentioning the structures that are relevant for the understanding of the paper. Also, the known role of Hdac1/2 in epidermal should be stated more clearly.*

We have discussed the findings of Ramsey et al., 2011 and the main results of the recent paper of LeBoeuf et al. in more detail at the end of the Introduction section.

- *Figure1:*

- *The IF stainings for Hdac1 and Hdac2 in Fig1 and Fig S1 should be co-stained with a basal marker such as K14 or β 2;4 and a suprabasal maker to allow for a better interpretation of the expression pattern of Hdac1/2.*

As suggested by the reviewer we have co-stained the sections in addition to HDAC1 and HDAC2 also with antibodies for K1 and K14 to better compare HDAC1/HDAC2 expression patterns (shown in Suppl. Figure S1A and B) and we detect both deacetylases in the K14- and K1-expressing layers.

- *The age of all mice shown (Fig1A, SFig2G) as well as the age of the mice for the skin samples (Fig1B, FigS2A,B,H) should be indicated in all panels. This should be done for all mice shown and all skin histology panels in all subsequent figures.*

The age of the respective mice is now indicated either on the figure or in the legend in all figures showing pictures with mice or histological sections.

- *The H&E staining or the IF presented in FigS2A and B are more informative of the correct development of the epidermis in Hdac1cKO and Hdac2cKO and should be included in Fig1 or replace Fig1A.*

Due to the restructuring of the manuscript as suggested by the reviewers we have moved now all

developmental *Hdac1cKO* and *Hdac2cKO* data to the Supplementary Figures and show the respective H&E staining as Suppl. Fig. 1G.

- *The methodology to quantify Ki67 positive cells (FigS2C) and epidermal thickness (FigS2D) should be clearly described in the materials and methods and statistical analysis should be preformed for all quantifications and the P values should be indicated in the figure legends. This should be done for all quantifications in all subsequent figures.*

- Statistical analysis should be preformed for the quantification of Fig1C and D and the P values should be indicated in the figure legends, including for FigS2E,D.

Where applicable the p-values are now shown as asterisk on the figures. For better visibility we indicate the p-values as numbers in the corresponding figure legends.

We show the relative increase in protein levels compared to wildtype control as mean value with corresponding s.d. The analysis for many proteins and the HDAC activity assays have to be performed immediately after isolation of the mouse tissue. Thus, these experiments cannot be performed at the same time but are data sets collected over months. According to our statisticians it is incorrect to combine data obtained from different immunoblots to calculate p-values.

Therefore we have stated in the Materials and Methods section:

If data were obtained from independent experiments, controls were set to 1 or 100% and the s.d. of the ratio of the mutant mice in relation to the controls is shown.

The same argument is valid for the quantifications shown in Figures 1F, 4D, and 5D,F,H,I as well in the Suppl. Fig. S1J, S2C, S6B,D.

The same argument is valid for the quantifications shown in Figures 1F, 4D, and 5D,F,H,I.

- *Figure2:*

- *Again, the IF stainings for Hdac1 and Hdac2 in Fig2B should be co-stained with basal and suprabasal makers to allow for a better interpretation of the expression pattern of Hdac1/2.*

See comments for Figure 1.

- *Statistical analysis should be preformed for the quantification of Fig2C and D and Fig24A,B and the P values should be indicated in the figure legends.*

Quantifications were performed and p-values are shown in Fig. 1G.

- *To fundament their claim that certain histone acetylation marks are increased in the epidermis of Hdac1cKO Hdac2het, the authors should quantify the immunoblots of FigS4C and perform statistical analysis.*

We have quantified the histone immunoblots for different histone acetylation marks. We show the relative increase in acetylation signals compared to wildtype control as mean value with corresponding s.d. The analysis for many proteins and the HDAC activity assays have to be performed immediately after isolation of the mouse tissue. Thus, these experiments cannot be performed at the same time but are dat sets collected over months. According to our statisticians it is incorrect to combine data obtained from different immunoblots to calculate p-values. The same argument is valid for the quantifications shown in Figures 1F, 4D, and 5D,F,H,I.

- *Figure3:*

- *The age of the mice analyzed in Fig3A, B and D should be indicated in the panels.*

Done, see same point under Figure 1.

- *The claim that there is an enlargement of the basal and spinous layer in the Hdac1cKO Hdac2het is not very clearly illustrated in Fig3B and the authors should present a more convincing example. Again, the description of how the quantification on the thickness of the epidermis was performed (Fig3C) should be clearly described.*

We have provided a detailed description of the method and the quantification in the Supplementary Information.

- *The graphs presented, such as in Fig3E should clearly indicated whether it is protein levels of mRNA that is being shown, making the interpretation of the figures more immediate. This is should be indicated in all graphs showing either protein levels or mRNA expression levels in all figures*

We fully agree with the reviewer and have inserted the corresponding headers in Figures 1, 2, 3, 4 and 5 and in all Supplementary Figs.

- *Figure4:*

- *Again, the authors should include the age of all samples shown in Fig4 as well as clear description of how the SG gland sizes was quantified and the P values for the statistical tests.*

Age and p-values are now shown in Figure 3 and the corresponding figure legends.

-*Figure5:*

- *The age of the skin analyzed for proliferation in Fig5A and S4E is not indicated. Also, the phenotype of the Hdac1cKO Hdac2het mice appears to become more severe with time (Fig S3D-H). In Hdac1/2 2KO there is impairment of barrier function of the skin. The data presented in this paper does not indicate whether there is also impairment of barrier function in the Hdac1cKO Hdac2het - this must be analyzed in the postnatal mice. Impairment of the barrier function of the skin has been shown to induce inflammation in the skin, which has been associated with increased proliferation and expression of differentiation markers. Since there is both an increase in proliferation but also an increase in expression of inflammation related makers such as K6, K16, S100a8 and S100a9A in the Hdac1cKO Hdac2het mice, the authors must address whether there is inflammation in these mutants. A time course analysis should be performed in neonatal mice to determine which defects can be detected earlier and whether there is an increase in hyperproliferation in the epidermis as the tissue begins to show more signs stress and damage. Also, the authors should investigate whether there is infiltration of immune cells in the epidermis with markers such as CD45.*

As suggested by the reviewer we have performed barrier function assays (TEWL and toluidine staining). As shown in Suppl. Figs. S4A and C there is no barrier defect observable in *Hdac1^{Δ/Δep} Hdac2^{Δ/+ep}* mice. In addition the expression of tight junction genes such as Occludin, Claudin 3 and Claudin 8 shows no significant difference in control and *Hdac1^{Δ/Δep} Hdac2^{Δ/+ep}* mice at P5 (Suppl. Fig. S4B).

We also analyzed a potential infiltration of immune cells by FACS analysis at P5, when proliferation is already increased in *Hdac1^{Δ/Δep} Hdac2^{Δ/+ep}* epidermis (Fig. 3H). As shown in Suppl. Fig. 4D and E FACS analysis for markers of immune cells at P5 indicated no significant increase in immune infiltrates in the mutant epidermis.

- *The authors should inhibit inflammation with the anti-inflammatory drug dexamethasone (Perez-Moreno et al, 2006) to address which aspects of the Hdac1cKO Hdac2het phenotype is due to inflammation and which might be due to increased cMyc protein.*

Furthermore, we performed dexamethasone experiments to determine whether a possible immune cell infiltration in the epidermis might impact the phenotype of the *Hdac1^{Δ/Δep} Hdac2^{Δ/+ep}* mice. We treated P4 mice for 30 days with 0.05 mg/kg dexamethasone by intraperitoneal injection. The phenotypic differences between the littermates correspond to normally observed pleiotropic variants. The expression of inflammation-related genes (Ccl5, Cxcl10, TNF-alpha, Traf6 and Cxcl1) is reduced upon Dexamethasone treatment in *Hdac1^{Δ/Δep} Hdac2^{Δ/+ep}* (The direct HDAC1 target gene Ada and innate immune gene Defb4 served as negative controls and remained unchanged during the Dexamethasone treatment). This confirms the effect of the anti-inflammatory drug and the lack of an effect on the developmental phenotype of mutant mice. In line with the barrier function assays and the FACS analysis described above we conclude that the phenotype of mutant mice is not linked to barrier function defects or immune infiltrates.

Dexamethasone treatment in *Hdac1 Δ/Δ ep Hdac2 $\Delta/+ep$* mice and control littermates

Figure legend: *Hdac1 Δ/Δ ep Hdac2 $\Delta/+ep$* mice are indicated with an asterisk

- The claim that *Hdac1*KO *Hdac2*het mice have defects in lineage determination and that there is a mobilization of epidermal stem cells should be carefully discussed. The data presented does not exclude that there is mobilization of the HF stem cell to the interfollicular epidermis as a result of wounding of the epidermis.

We cannot fully exclude this possibility and discuss this on page 19. It is not likely that wounding induces mobilization of HF stem cells at the back skin, since wounding in mutant mice is restricted to the tail and not detectable on most areas of the back skin and we do not detect immune infiltrates in the mutant skin.

- The methodology for quantifying the signal from *in situ* hybridization and the BrdU⁺ cells in the epidermis should be described clearly in the materials and methods.

We have provided a detailed description of the method in the Supplementary Information.

- The expression level of cMyc immunoblot (Fig5G) should be quantified.

The quantification is shown in Figure 4D.

- Figure6:

- What are the 79 genes that are affected in the *Hdac1*KO and that cannot be compensated by *Hdac2*? Are these relevant for the interpretation of the phenotype of *Hdac1*KO *Hdac2*het versus the *Hdac1/2* 2cKO? Are these genes related to inflammation? Also, does *Hdac1* bind to the promoter of these genes directly? This could be tested with the ChIP already performed and could be very informative on why the *Hdac1*KO *Hdac2*het and the *Hdac1/2* 2cKO mice have different phenotypes.

We have previously shown that genes that are deregulated upon loss of HDAC1 have less compensating HDAC2 bound to the promoter, whereas other genes are not deregulated in the absence of HDAC1 due to increased recruitment of HDAC2 (Zupkovitz et al. 2006). This is also true for the genes shown in Figure 6 and Suppl. Fig. S7. As discussed above Reviewer 1, Point 14 the presence of HDAC1 at a promoter tells little about a potential regulation of the respective gene. Therefore we decided to include Sin3A and histone acetylation ChIPs in our analysis of target genes (see below).

- The effect of loss of deacetylase activity in *Hdac1cKO Hdac2het* mice appears to only have an indirect effect on c-Myc or at least only affect its post-translational stability. What is the nature of the functional link between Sin3A and cMyc? Could the instability of the *Hdac1* repressor complex with Sin3A and MTA2 explain the persistence of cMyc protein? Does cMyc interact directly with *Hdac1* repressive complexes? The authors should better discuss their hypothesis.

In addition to their chromatin related role as histone deacetylases HDACs have also an important function as deacetylases of non-histone substrates. Our data in combination with the previous findings of Nascimento et al., 2011 reveal a novel and important role of HDAC1 as direct regulator of the c-Myc protein. Nascimento et al. have shown that Sin3A directly interacts with c-Myc and when overexpressed destabilizes the c-Myc protein. We show that reduced activity of the Sin3A complex in the absence of HDAC1 and one allele of *Hdac2* results in increased protein levels. HDAC inhibition with an inhibitor with preference for HDAC1 leads to stabilization of c-Myc (Suppl. Fig. S6D) indicating that HDAC1 is the relevant catalytic subunit of the Sin3A complex for the attenuation of c-Myc protein levels in the epidermis. Based on the reviewer comment we have discussed the model in detail in the Discussion section (page 19).

- The immunoblots in Fig6 E and G should be quantified to support the claims made by the authors on the change in the levels of Sin3A and MTA2.

The quantification is shown in Figures 5 H and I.

- Figure7:

- How generalizable is the absence of complete compensation by *Hdac2* in the *Hdac1cKO Hdac2het* mice? The authors only show two genes (*Spr2h* and *Epgn*) in Fig7B. It will be important to understand whether other genes behave like this. The authors should present other examples of genes that have *Hdac1* directly bound to their promoter. Furthermore, might any of the 79 genes that are not at all compensated by *Hdac2* in the *Hdac1cKO* play a role in the phenotype observed? While the deregulation alone of these genes might not have a phenotype, they could interact with other genes that are deregulated in the *Hdac1cKO Hdac2het* mice. This might be an interesting point for discussion. Finally, can the authors comment on how many of the deregulated genes have *Hdac1* directly bound at their promoter? This could help explain direct and indirect effects of loss of HDacs and therefore their role in promoting hyperproliferation or suppressing it, depending on the context.

As suggested by the reviewer we have performed ChIP experiments for more genes: We have included the hair follicle regulators *Hoxc13* and *Msx2* the epidermis markers *Spr2h*, *S100a8*, *S100a9*, *Lce3b*, *Epgn*, *Ada* and the SG marker *Klk6* in our ChIP assays. To understand the biological consequence of loss of HDAC1 and reduction of HDAC2 on these genes we have performed in addition to HDAC1, HDAC2 ChIPs also histone acetylation ChIPs (Figure 6 and Suppl. Fig. S7). Given that 79 genes are deregulated in the *Hdac1^{Δ/Δep}* epidermis and 3749 in *Hdac1^{Δ/Δep} Hdac2^{Δ/+ep}* epidermis it is difficult to judge whether some of the 79 genes contribute in addition to the phenotype of *Hdac1^{Δ/Δep} Hdac2^{Δ/+ep}* mice.

-Figure8

- In continuation of the comment above, the authors should discuss why *Hdac1* might have a tumor suppressor role and why *Hdac2* does not. What are the relevant targets that might justify this difference?

As described in more detail in the revised version the main target is c-Myc. Loss of Sin3A co-repressor function in HDAC1-deficient SOS tumors results in increased protein levels of c-Myc and the c-Myc target *Skp2*, hyperproliferation and accelerated tumor development. In contrast, loss of HDAC2 in SOS tumors does not impair co-repressor functions. Consequently, c-Myc is not induced and proliferation and tumor development is unchanged. These data are shown now in Figures 7 and 8.

- Inflammation has been shown to have a link with tumor development. The authors must inhibit inflammation in their K5-SOS *Hdac1cKO* mice to address whether *Hdac1* has a tumor suppressor independent of inflammation.

HDAC1-deficient SOS tumor mice are moribund and have to be sacrificed some days after birth. Based on the resulting low number of SOS tumor mice and that there is no evidence for immune infiltration in the absence of HDAC1 we performed analyses of several immune genes in *Hdac2^{Δ/Δep}* and *Hdac1^{Δ/Δep}* SOS tumors (n=3) and the corresponding control tumors. As shown in the attached

Figure we did not observe of significant immune response in *Hdac1*^{Δ/Δep} SOS mice. This indicates that the accelerated and increased tumor formation in *Hdac1*^{Δ/Δep} SOS mice is not due to the slight, albeit not significant immune response.

RNA expression

- Again, information on age of the mice is missing from the figure, as well as quantification for some the immunoblots and statistics.

Age and quantifications were added, statistics where appropriate.

Minor comments:

- Fig3A has a white arrow that is not described in the figure legend neither in the text.

The arrows have been removed.

Thank you very much for the revised study that has been assessed by two of the original referees

I kindly ask you to consider and integrate the few minor remarks from one of the original referees. Please provide us with an ultimate text-file for publication via reply E-mail.

Please also notice that The EMBO Journal encourages the publication of source data, particularly for electrophoretic gels/blots, with the aim to make primary data more accessible and transparent to the reader. This entails presentation of un-cropped/unprocessed scans for KEY data of published work. We would be grateful for one PDF-file PER FIGURE with such information. These will be linked online as supplementary "Source Data" files.

I like to congratulate you at this stage to the study and look forward to receiving relevant source data as soon as possible to facilitate production/publication still in 2013.

Be assured that the editorial office will be in touch soon after receipt of the requested files/information to formally accept the paper.

REFeree REPORTS:

Referee #1:

The authors have addressed all my initial concerns and the quality of the manuscript improved considerably with the new data. Below are my largely minor comments on the revised version on the manuscript.

- (1) Page 7 line 1-3: The claim that HDAC1-deficient epidermis is higher sensitive towards mechanical stress is not supported by any experimental evidence and should be deleted.
- (2) Page 8 last sentence: refer to figure 2C
- (3) Page 9 line second sentence: refer to figure 2D
- (4) The authors have not addressed my concern regarding the quantification of quiescent cells by measuring LCRs. Figure 3H shows that after one pulse of BrdU almost twice as many cells incorporated the label. If twice as many cells are labeled at the pulse, a 2-fold increase of BrdU-positive cells after the chase (Figure 4B) does not seem surprising and does not seem to allow conclusions regarding slow cycling populations. What is the percentage of BrdU-labeled cells at the pulse (at P6 pulsed for 72h) in Hdac1^{-/-} Hdac2^{+/-} versus the control?
- (5) Figure 4B: The authors should clarify in the text that the loss of the BrdU label in Hdac1^{-/-} Hdac2^{+/-} hair follicles is in this particular case the very likely result of loss of cells due to apoptosis as opposed to loss of the label due to cycling. Otherwise, the data are at odds with figures 2D and 3H.
- (6) Page 12: The two sentences: "These findings reveal profound changes in epidermal lineage commitment..." and "The increased proliferation in IFE and SG..." are not supported by direct evidence in the manuscript. Such claims would require lineage tracing experiments and the claims needs to be toned down.
- (7) Figure 5C and header on page 14: the label Hdac1^{+/-ep} Hdac2^{-/-} needs correction.

Referee #2:

The authors addressed all our comments. The paper can be accepted for publication.

Thank you for the good news. Please find attached the final manuscript file. We have integrated the comments of Reviewer 1 as follows.

- (1) Page 7 line 1-3: The claim that HDAC1-deficient epidermis is higher sensitive towards mechanical stress is not supported by any experimental evidence and should be deleted. We have deleted the sentence referring to the increased sensitivity towards mechanical stress.
- (2) Page 8 last sentence: refer to figure 2C
Done.
- (3) Page 9 line second sentence: refer to figure 2D
Done.
- (4) The authors have not addressed my concern regarding the quantification of quiescent cells by measuring LRCs. Figure 3H shows that after one pulse of BrdU almost twice as many cells incorporated the label. If twice as many cells are labeled at the pulse, a 2-fold increase of BrdU-positive cells after the chase (Figure 4B) does not seem surprising and does not seem to allow conclusions regarding slow cycling populations. What is the percentage of BrdU-labeled cells at the pulse (at P6 pulsed for 72h) in *Hdac1^{-/-} Hdac2^{+/-}* versus the control?
As suggested by the reviewer we have performed pulse experiments showing increased BrdU incorporation in the mutant IFE (Fig. 3H). We cannot exclude that the increased number of LRCs is at least in part due to a higher number of labeled progenitor cells in the mutant IFE. Therefore we have toned down the conclusions on page 12 and have also discussed the alternative scenario of increased apoptosis in hair follicles and increased proliferation in the IFE of mutant mice.
- (5) Figure 4B: The authors should clarify in the text that the loss of the BrdU label in *Hdac1^{-/-} Hdac2^{+/-}* hair follicles is in this particular case the very likely result of loss of cells due to apoptosis as opposed to loss of the label due to cycling. Otherwise, the data are at odds with figures 2D and 3H.
We have added a corresponding sentence with reference to Figure 2D on page 12.
- (6) Page 12: The two sentences: "These findings reveal profound changes in epidermal lineage commitment...." and "The increased proliferation in IFE and SG..." are not supported by direct evidence in the manuscript. Such claims would require lineage tracing experiments and the claims needs to be toned down.
Hyperproliferation of the IFE is demonstrated in Fig. 3I. As suggested we modified the statements concerning the SG proliferation and the lineage commitment.
- (7) Figure 5C and header on page 14: the label *Hdac1^{+/-ep} Hdac2^{-/-}* needs correction.
We have corrected the header in the Results section (page 14) and need to up-load a corrected version of Figure 5.

I hope that the manuscript is now suitable for publication in the EMBO Journal.